# Locomotor deficits in a mouse model of ALS are paralleled by loss of V1-interneuron connections onto fast motor neurons

Ilary Allodi 🄲 [1,4✉], Roser Montañana-Rosell 🄲 [1,4], Raghavendra Selvan 🄲 [1,2], Peter Löw[3] & Ole Kiehn 🄲 [1,3✉]

ALS is characterized by progressive inability to execute movements. Motor neurons inner-vating fast-twitch muscle-fibers preferentially degenerate. The reason for this differential vulnerability and its consequences on motor output is not known. Here, we uncover that fast motor neurons receive stronger inhibitory synaptic inputs than slow motor neurons, and disease progression in the SOD1[G93A] mouse model leads to specific loss of inhibitory synapses onto fast motor neurons. Inhibitory V1 interneurons show similar innervation pat-tern and loss of synapses. Moreover, from postnatal day 63, there is a loss of V1 interneurons in the SOD1[G93A] mouse. The V1 interneuron degeneration appears before motor neuron death and is paralleled by the development of a specific locomotor deficit affecting speed and limb coordination. This distinct ALS-induced locomotor deficit is phenocopied in wild-type mice but not in SOD1[G93A] mice after appearing of the locomotor phenotype when V1 spinal interneurons are silenced. Our study identifies a potential source of non-autonomous motor neuronal vulnerability in ALS and links ALS-induced changes in locomotor phenotype to inhibitory V1-interneurons.

[1] Department of Neuroscience, Faculty of Health and Medical Sciences, University of Copenhagen, Copenhagen N, Denmark. [2] Department of Computer Science, University of Copenhagen, Copenhagen N, Denmark. [3] Department of Neuroscience, Karolinska Institutet, Stockholm, Sweden. [4] These authors contributed equally: Ilary Allodi, Roser Montañana-Rosell. ✉email: iallodi@sund.ku.dk; Ole.Kiehn@sund.ku.dk

Amyotrophic lateral sclerosis (ALS) is a neurodegenerative disorder characterized by a progressive inability to execute movements including breathing. During the progression of the disease, somatic motor neurons in the spinal cord and brain stem, as well as corticospinal neurons in the motor cortex degenerate[1,2]. The degeneration of motor neurons is the direct cause of the paralysis. However, motor neurons that innervate different muscle fibers are not equally affected in ALS. Specifically, motor neurons innervating fast-twitch fatigable fibers—those that produce strong force during movement—are more vulnerable to ALS degeneration than motor neurons that innervate slow-twitch or fast fatigable resistant fibers controlling sustained muscle contraction[2–5]. Consequently, motor neurons innervating fast-twitch fatigable fibers are first affected in ALS. The reason for this differential vulnerability and its consequences for motor output is not known.

While the etiology of most cases of ALS is unknown, mutations in the SOD1 gene are linked to some familial forms. The SOD1-induced motor neuron degeneration has been linked to motor neuron intrinsic toxicity but also to SOD1-mediated extrinsic factors, including changes in nearby astrocytes and microglia as well as nerve ensheathing Schawnn cells[6,7]. In the present study we investigate if the differential vulnerability may be linked to non-motor neuron autonomous circuits in the spinal cord. One such source may be a change in the excitatory or inhibitory synaptic interneuron inputs to motor neurons[8]. Previous studies have reported inhibitory dysfunctions in an ALS SOD1$^{G93A}$ mouse model demonstrating loss of glycinergic, but not GABAergic innervation of limb motor neurons[9–11] and reduced glycine receptor in the spinal cord in ALS[12]. Reduction of inhibitory inputs was also seen in a SOD1 zebrafish model[13].

The glycinergic synaptic inputs to motor neurons in the spinal cord mostly originate from inhibitory interneurons in the ventral spinal cord, including Ia inhibitory neurons (reciprocal inhibition), Renshaw Cells (recurrent inhibition), and commissural interneurons (crossed inhibition among others)[14,15]. The inhibitory interneurons of the ventral spinal cord can be classified into three major classes of neurons characterized by their expression of transcription factors: V0 neurons expressing developing brain homeobox protein 1 (Dbx1), V1 neurons expressing Engrailed 1 (En1), and V2b neurons expressing Gata2/3. Selective ablation of these different interneuron classes leads to specific motor defects, with loss of left-right coordination at slow speed of locomotion with loss of inhibitory V0 neurons[16–18], slowing of locomotor speed with loss of all V1 neurons[19], reduction in intralimb extension with loss of V1 neurons or loss of flexor-extensor alternation when both V1 and V2b neurons are ablated from the spinal cord[20,21]. The En1 transcription factor defines the class of V1 interneurons during development[22,23], postnatally[24–26] and in adult mice[20,21] and can therefore be used as a marker to target this group of interneurons.

In the present study, we found (1) that motor neurons innervating fast-twitch fatigable fibers receive a stronger glycinergic input than motor neurons innervating slow and fast non-fatigable fibers, (2) that there is a selective loss of glycinergic synapses on fast motor neurons during disease progression in an ALS SOD1$^{G93A}$ mouse model, (3) that En1-positive synapses follow similar innervation pattern on motor neurons under normal physiological conditions and similar loss of connectivity during ALS development as the glycinergic population, (4) that the number of En1 positive neurons is reduced during ALS development and that the reduction happens before motor neuron degeneration, (5) that the loss of glycinergic and V1 synapses on motor neurons is paralleled by a progressive slowing of locomotor speed, capability to maintain locomotion and distinct changes in intralimb coordination that precede motor neuron degeneration,

(6) and that reversible silencing of spinal En1 neurons phenocopy the ALS-induced locomotor phenotype in wild-type mice but has no effect in ALS mice. These findings suggest an inhibitory interneuron contribution to motor neuron vulnerability in ALS and demonstrate that this motor neuron-specific defect leads to locomotor changes that appear before motor neuron degeneration and is linked to En1 inhibitory interneurons.

## Results

**Fast motor neurons receive stronger inhibitory glycinergic connections than slow motor neurons.** To evaluate the inhibitory connections to fast and slow motor neurons we used a GlyT2$^{GFP}$ (glycinergic transporter 2) mouse line[27] where eGFP reliably labels glycinergic neurons and their terminals in the spinal cord[28]. Previous studies have shown that fast and slow motor neurons can be visualized in the spinal cord by the presence of specific markers among them matrix metalloproteinase 9 (MMP9) is a marker for fast motor neurons[29–34], and estrogen-related receptor beta (ErrBeta) that is marker for the slow ones[35,36]. Thus, fast motor neuron somata in the ventral horn were labeled with antibody against MMP9 (Fig. 1a) and slow motor neurons were labeled with antibody against ErrBeta (Fig. 1e). Only large cells (over 200 μm$^2$ in soma area) present in the ventral horn of the spinal cord were quantified and the synaptic vesicle marker synaptophysin (SYN) was used to mark putative synaptic terminals[37] (Fig. 1c, g). The GFP positive terminals around somata were specifically selected (Fig. 1b, f). To account for differences in somata size between slow and fast motor neurons[38] all measurements were normalized for the soma areas and expressed as densities (Fig. 1d, h, q–r). A minimum of 200 motor neurons from the lumbar segment of the spinal cord were quantified per condition. The overall synaptic density was larger on fast motor neurons than on slow motor neurons (MMP9 = 566 ± 60.75 and ErrBeta = 417.19 ± 17.5; $t$ test, $P$ = 0.0317; $N$ = 9) (Fig. 1q) and glycinergic terminals were also found at higher density around fast motor neuron somata (MMP9 = 274.4 ± 29.25 and ErrBeta = 142.75 ± 39.77; $t$ test, $P$ = 0.0008; $N$ = 9) (Fig. 1r).

**Fast motor neurons lose inhibitory glycinergic connections before slow motor neurons.** We next crossed Glyt2$^{GFP}$ mice with SOD1$^{G93A}$ mice and performed measurements of SYN and eGFP density on MMP9$^+$ and ErrBeta$^+$ motor neurons (Fig. 1i-p, s–t) at postnatal (P) day 45, 63, and 84. The genotype of the newly derived SOD1$^{G93A}$;Glyt2$^{GFP}$ mice were assessed for copy number of the human SOD1$^{G93A}$ mutated transgene that was the same in SOD1$^{G93A}$;Glyt2$^{GFP}$ and pure SOD1$^{G93A}$ mice (Supplementary Fig. 1d). The change in weight in the two strains (SOD1$^{G93A}$ Fig. 2a and SOD1$^{G93A}$;Glyt2$^{GFP}$ Supplementary Fig. 1a) as well as the survival end point were similar (SOD1$^{G93A}$;Glyt2$^{GFP}$ survival = 157.6 ± 7.2 days, $N$ = 8; SOD1$^{G93A}$ survival = 161.6 ± 5.3 days, $N$ = 11) (Supplementary Fig. 1c). Moreover, motor neurons (Supplementary Fig. 2a–c) were found in the same quantity as in previous reports of the original SOD1$^{G93A}$ mouse model[39,40] with a reduced number of motor neurons present at P112 (Supplementary Fig. 2c). Thus, the SOD1$^{G93A}$;Glyt2$^{GFP}$ mice showed no differences from the original SOD1$^{G93A}$ strain with respect to ALS phenotype. When analyzing the synaptic density at postnatal (P) day 63 we found that there was a reduction to around 50% of SYN inputs (one-way ANOVA and Dunnett's post hoc, $P$ = 0.0382; $N$ = 3) (Fig. 1s) onto MMP9$^+$ motor neurons, while eGFP positive inputs were already reduced to 51% at P45 when compared to wild-type mice (one-way ANOVA and Dunnett's post hoc, $P$ = 0.0121; $N$ = 3) (Fig. 1t). This reduction onto MMP9$^+$ motor neurons continued dramatically with ALS progression and was down to 9.94% at P84 compared to age-matched wild-

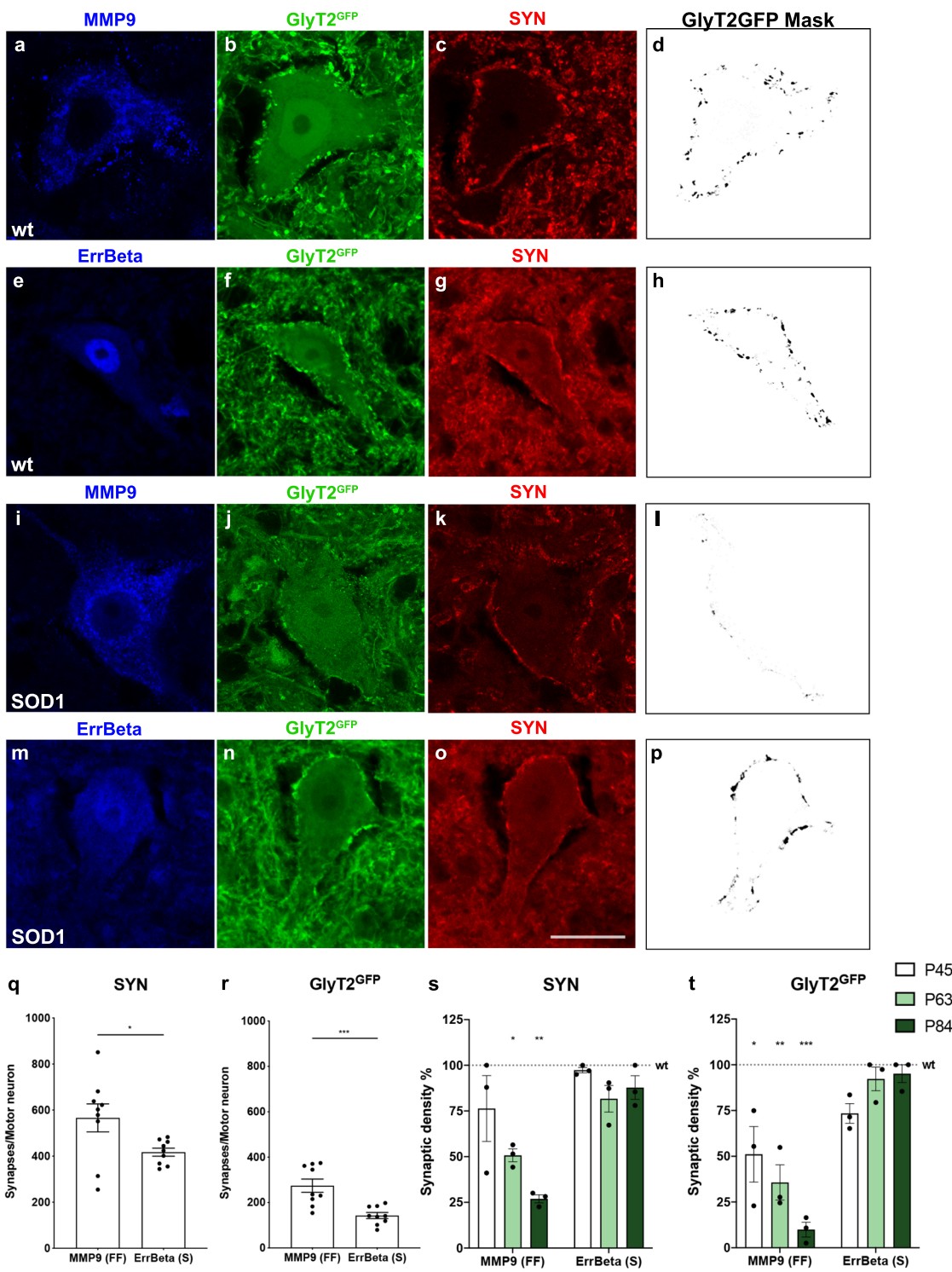

type controls (one-way ANOVA and Dunnett's post hoc, $P = 0.00007$; $N = 3$) (Fig. 1t). The reduction of eGFP$^+$ terminals on fast motor neurons at these timepoints was not due to loss of motor neurons or shrinkage in their size (Supplementary Fig. 2a–d). Motor neuron loss was observed at later timepoints[38,41,42] (Supplementary Fig. 2c). Overall, the general synaptic loss on MMP9$^+$ motor neurons —as measured with SYN immunoreactivity—seemed to be less than the reduction in glycinergic terminals, suggesting that glycinergic synapses might be primarily affected (Fig. 1s-t). In contrast to these changes, the analysis conducted on ErrBeta$^+$ motor neurons showed that glycinergic inputs on slow motor neurons are intact at P84 and

comparable to what is seen in age-matched wild-type littermates (Fig. 1s–t). Moreover, glutamatergic inputs onto fast and slow motor neurons were assessed at P45 and P63 timepoints utilizing a Vglut2 (vesicular glutamate 2) antibody (Supplementary Fig. 2e–j). Vglut2 is expressed in all glutamatergic interneurons in the ventral spinal cord[18,43]. Staining was performed in the SOD1$^{G93A}$;Glyt2$^{GFP}$ wild-type and transgenic littermates negative for eGFP while SYN antibody was used as counterstaining (Supplementary Fig. 2e–h). At these timepoints there were no significant changes in the synaptic density of soma-near Vglut2 buttons onto MMP9$^+$ and ErrBeta$^+$ motor neurons in SOD1$^{G93A}$;Glyt2$^{GFP}$ mice as compared to wild-

**Fig. 1 Preferential glycinergic innervation of motor neurons innervating fast-twitch fatigable muscle fibers and loss in a SOD1$^{G93A}$ mouse model.**
Microphotographs depicting glycinergic inputs onto MMP9$^+$ fast motor neurons (**a**–**c**) and ErrBeta$^+$ slow motor neurons (**e**–**g**) in a GlyT2$^{GFP}$ mouse. Motor neuron somata were selected and synapses were reconstructed for quantifications as shown in the masks in (**d**) and (**h**). Quantifications (**q**–**r**) expressed as synaptic particles and corrected for motor neuron area, show higher synaptic density in MMP9$^+$ motor neurons for both synaptophysin (SYN) (MMP9 = 566 ± 60.75 and ErrBeta = 417.19 ± 17.5; two-tailed $t$ test, $P = 0.0317$, $t = 2.354$, df = 16; $N = 9$ independent mice) and GlyT2$^{GFP}$ (MMP9 = 274.4 ± 29.25 and ErrBeta = 142.75 ± 39.77; two-tailed $t$ test, $P = 0.0008$, $t = 4.100$, df = 16; $N = 9$ independent mice) markers. Innervation of fast (**i**–**k**) and slow (**m**–**o**) motor neurons in GlyT2$^{GFP}$ mice crossed with SOD1$^{G93A}$ mice at postnatal day (P) 63 and their respective synaptic densities (**l**, **p**). Quantifications of synaptic density were performed at P45, P63, and P84 for SYN (**s**) and GlyT2$^{GFP}$ (**t**) markers. MMP9$^+$ motor neurons showed progressive reduction of SYN$^+$ (one-way ANOVA and Dunnett's post hoc, $F(3,14) = 7.004$, P45 $P = 0.6992$, P63 $P = 0.0382$, P84 $P = 0.0027$; $N = 3$ independent mice per timepoint) and GlyT2$^{GFP+}$ inputs (one-way ANOVA and Dunnett's post hoc, $F(3, 14) = 16.742$, P45 $P = 0.0121$, P63 $P = 0.0015$, P84 $P = 0.00007$; $N = 3$ independent mice per timepoint) when compared with ErrBeta$^+$ motor neurons (SYN$^+$ one-way ANOVA and Dunnett's post hoc, $F(3, 14) = 2.224$, P45 $P = 0.7919$, P63 $P = 0.1285$, P84 $P = 0.9495$; $N = 3$ independent mice per timepoint) (GlyT2$^{GFP+}$ one-way ANOVA and Dunnett's post hoc, $F(3, 14) = 2.732$, P45 $P = 0.057$, P63 $P = 0.9999$, P84 $P = 0.9694$; $N = 3$ independent mice per timepoint). MMP9 (**a**, **i**) or ErrBeta (**e**, **m**) in blue, SYN (**c**, **g**, **k**, **o**) in red and GlyT2$^{GFP}$ (**b**, **f**, **j**, **n**) in green. Scale bar in (**o**) = 50 μm representative for all images. A minimum of 200 motor neurons was quantified per condition. In all graphs, data are presented as mean values ± SEM. Source data are provided as a Source Data file.

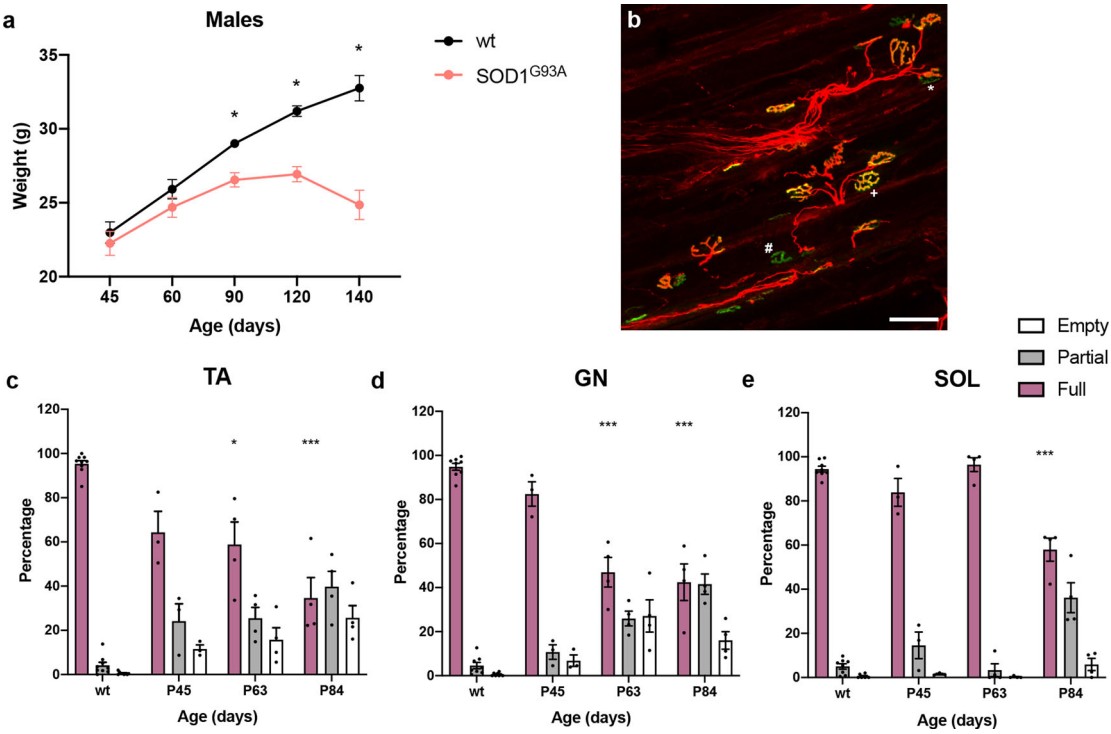

**Fig. 2 Neuromuscular junction (NMJ) innervation assessment in the fast-twitch fatigable *Tibialis Anterior* (TA) (c) and *Gastrocnemius* (GN) (d) muscles and in the slow-twitch *Soleus* (SOL) (e) muscle. a Weights of the SOD1$^{G93A}$ mice included in the study and used for comparison with the multiple crossing conditions.** Differences in weights are observed from postnatal day 90 (two-tailed multiple $t$ tests, P90 $P = 0.00008$, $t = 4.773$, df = 24; P120 $P = 0.0000003$, $t = 7.125$, df = 23; P140 $P = 0.0012$, $t = 5.713$, df = 6; $N = 15$ independent mice per condition). NMJ were categorized either as fully innervated, shown by + in (**b**), or as partially innervated, shown as * in (**b**), or as empty, shown as # in (**b**). Significant denervation in TA (**c**) (one-way ANOVA and Dunnett's post hoc, $F(3, 12) = 9.767$, P63 $P = 0.0200$, P84 $P = 0.0005$) and GN (**d**) was in P63 mice (one-way ANOVA and Dunnett's post hoc, $F(3, 12) = 19.36$, P63 $P = 0.0003$, P84 $P = 0.0001$). **e** In SOL, significant denervation was much later starting from P84 (one-way ANOVA and Dunnett's post hoc, $F(3, 12) = 20.82$, P84 $P = 0.00005$) (wt $N = 8$, SOD1$^{G93A}$ P45 $N = 3$, P63 $N = 4$, P84 $N = 4$ independent mice). Scale bar in (**b**) = 50 μm. A minimum of 800 NMJs were analyzed per condition. In all graphs, data are presented as mean values ± SEM. Source data are provided as a Source Data file.

type littermates (Supplementary Fig. 2i–j). Together these data demonstrate that MMP9$^+$ neurons receive stronger inhibitory inputs than ErrBeta$^+$ motor neurons and that there is a specific soma-near synaptic loss of inhibitory synapses on MMP9$^+$ motor neurons that starts before the somatic spinal motor neurons are affected in SOD1$^{G93A}$-induced ALS, while glutamatergic inputs are not altered at early disease stages.

Furthermore, the changes in inhibitory inputs to motor neurons preceded changes of synapses at the neuromuscular junctions (NMJs). NMJs in fast-twitch fatigable muscles were first affected around P63 (Fig. 2b–e). The number of fully innervated endplates was significantly reduced at P63 to 58.84% of control in the fast *tibialis anterior* (one-way ANOVA and Dunnett's post hoc, P63 $P = 0.0200$) and 46.94% in the *gastrocnemius* muscles (one-way ANOVA and Dunnett's post hoc, P63 $P = 0.0003$) (over 800 NMJs were analyzed per condition at each timepoint and 3–4 animals per timepoint). In contrast the *soleus* muscle —composed of 50% slow-twitch fatigable resistant and 50% fast-twitch

fatigable resistant fibers—did not show reduction of fully innervated endplates until P84 (one-way ANOVA and Dunnett's post hoc, $P = 0.00005$) (Fig. 2c–e).

**En1 inhibitory spinal interneurons have stronger inputs on fast motor neurons than slow and show selective loss in SOD1$^{G93A}$ induced degeneration**. We next tested if the V1 subpopulation of spinal inhibitory neurons was involved in the differential innervation of MMP9$^+$ and ErrBeta$^+$ motor neurons and in the degeneration-induced synaptic stripping. The En1 transcription factor is found in ipsilaterally projecting neurons in the intermediate and ventral parts of the spinal cord which are known to project to motor neurons innervating the limbs with soma-near connections. These interneurons account for between 39 and 55% of inhibitory synapses found on motor neuron somata[20] and, in the adult mouse, 80% of these synapses are GlyT2 positive[44]. We used a viral approach to visualize the En1$^+$ terminals and cell bodies by injecting AAV1-phSyn1(S)-FLEX-tdTomato-T2A-SypEGFP-WPRE[45] (Fig. 3a) on each side of the lumbar (L) spinal cord at the L2-L3 level (two injections of 80 nl) in P42 En1$^{cre}$ mice[25,46] and P42 En1$^{cre}$ mice crossed with SOD1$^{G93A}$ mice. The genotype and phenotype of the cross for SOD1$^{G93A}$ did not differ from the non-crossed SOD1$^{G93A}$ strain and crossed mice reached the same end-stage at 157.7 ± 7.9 ($N = 6$) as the non-crossed SOD1$^{G93A}$ (Supplementary Fig. 1b–d). Since the viral approach only allows visualization of the synaptic inputs of the En1 neurons transduced by the virus, the total number of TdTomato positive cells per animal was quantified and used to normalize the synaptic density values. Synaptic density values were further normalized by motor neuron area to account for differences in cell size. In En1$^{cre}$ mice at P63 we found that the En1$^+$ soma-near synaptic terminal density (EGFP) was significantly stronger on fast motor neurons than on slow motor neurons (MMP9 = 49.2 ± 11.4 and ErrBeta = 33.2 ± 9; $t$ test, $P = 0.0401$; $N = 5$ – a minimum of 300 motor neurons were analyzed per condition) (Fig. 3b). In SOD1$^{G93A}$;En1$^{cre}$ mice at P63 the overall soma-near synaptic terminal density was visibly reduced as compared to En1$^{cre}$ mice (Supplementary Fig. 3a–b). Moreover, the number of TdT positive (En1) neurons was significantly lower in the SOD1$^{G93A}$;En1$^{cre}$ mice than in En1$^{cre}$ mice (En1$^{cre}$ = 99 ± 11.62%; SOD1$^{G93A}$;En1$^{cre}$ = 29.03 ± 13.41%; $t$ test, $P = 0.0040$; $N = 5$) (Supplementary Fig. 3c). The synaptic density analysis showed overall loss of soma-near En1$^+$ terminals on motor neurons in the SOD1$^{G93A}$;En1$^{cre}$ mice at P63 (Fig. 3c–l), with a dramatic reduction to 13% on fast motor neurons ($t$ test, $P = 0.00005$; $N = 5$) (Fig. 3c–g) and to 53% on the slow motor neurons compared with En1$^{cre}$ mice ($t$ test, $P = 0.0245$; $N = 5$) (Fig. 3h–l). These data show that inhibitory En1 interneurons have a strong innervation on MMP9$^+$ fast motor neurons and that these synapses are retracted or lost during the disease progression in SOD1$^{G93A}$ mice.

**The number of En1 positive neurons is reduced in SOD1$^{G93A}$ mouse compared to wild-type littermates before motor neuron degeneration**. To further investigate the fate of V1 interneurons in the SOD1$^{G93A}$ mouse and to evaluate if the loss of synapses could be attributed solely to a retraction of inputs or to a loss of V1 interneurons or both, we determined the number of En1$^+$ neurons in the spinal cord at different timepoints. For this we performed RNAscope in situ hybridization[47] to determine the En1 transcript in P45, P63, and P84 mice in wild-type littermates and SOD1$^{G93A}$ mice (Fig. 4a–c). In wild-type littermates there was on average 74.5 ± 7 En1$^+$ neurons per hemicord section ($N = 9$), while at P63 and P84 the number of cells was reduced to 54.8 ± 5.7 and 55.1 ± 1.6, respectively ($N = 3$, between 8–10

hemicord sections—12 μm thickness—were quantified per mouse). In SOD1$^{G93A}$ mice the number of En1 positive neurons did not differ at P45, but it decreased to 75% starting from P63 when compared to wild-type littermates. At P84 the number of En1$^+$ neurons was still reduced (one-way ANOVA and Dunnett's post hoc, P45 $P = 0.3108$, P63 $P = 0.0010$, P84 $P = 0.0003$; $N = 3$ per condition per timepoint) (Fig. 4d). However, when quantifying the amount of En1 transcripts within the analyzed cells, we could not detect a change in the level of expression between SOD1$^{G93A}$ and wild-type mice (two-way ANOVA and Tukey's post hoc, $P = 0.9729$; $N = 3$) (Fig. 4e). Together, these data show a loss of En1$^+$ interneurons at P63 in SOD1$^{G93A}$ which is maintained but not worsened during disease progression, and that the level of En1 transcript in surviving cells remains the same as in wild-type animals.

**SOD1$^{G93A}$ mice develop progressive slowing of locomotor speed at early timepoints while grip strength force is not reduced**. Fast and slow motor neurons are recruited in different motor tasks— with slow motor neurons recruited at low task forces and fast motor neurons at higher task forces. To evaluate the contribution of fast and slow motor neurons to motor output we assessed the locomotor performance of SOD1$^{G93A}$ mice over time. Moreover, we assessed the grip strength that mainly measures the contribution from low force contraction.

For locomotion, mice were tested weekly between P49 until P112 on a treadmill at a speed of 20 cm/s for 10 s in three consecutive sessions (corresponding to fast walk or trot[17]). Videos were tracked by DeepLabCut analysis[48] (Fig. 5a, Supplementary Movie 1). Over time SOD1$^{G93A}$ mice showed progressive incapability of coping with the 20 cm/s speed and increased dragging events (Fig. 5b, Supplementary Movie 2). To compensate for the reduced locomotor capability developing over time, the speed of the belt was decreased to either 15, 10, or 5 cm/s depending on the severity of the phenotype. The timepoint in which a SOD1$^{G93A}$ mouse could not sustain a speed of 20 cm/s was called Onset of locomotor phenotype. Notably, between P49 and P63 46.2% of the SOD1$^{G93A}$ mice were unable to follow the 20 cm/s treadmill speed (median = 77; Gehan-Breslow-Wilcoxon test, $P = 0.0003$; $N = 11$) (Fig. 5b; Supplementary Movie 1 and 2). This inability continued throughout the observation period. This is unlike age-matched control littermates that all were able to follow 20 cm/s treadmill speed (two-way ANOVA and Dunnett's post hoc, P63 $P = 0.0060$, P70 $P = 0.0085$, P77 $P = 0.0027$, P84 $P = 0.00009$; wt $N = 8$, SOD1$^{G93A}$ $N = 15$) (Fig. 5c). Over time, SOD1$^{G93A}$ mice also showed the decreased capability of accelerating while coping with the speed of the belt (two-way ANOVA and Dunnett's post hoc, $P = 0.0468$; wt $N = 8$, SOD1$^{G93A}$ $N = 15$) (Fig. 5d), decreased step frequency (two-way ANOVA and Dunnett's post hoc, $P = 0.0167$; wt $N = 8$, SOD1$^{G93A}$ $N = 15$) (Fig. 5f) and a tendency to make shorter steps (reduced stride length two-way ANOVA and Dunnett's post hoc, $P = 0.2157$; wt $N = 8$, SOD1$^{G93A}$ $N = 15$) (Fig. 5e) compared to wild-type littermates. At the Onset of locomotor phenotype only 50% of the SOD1$^{G93A}$ mice could follow a speed of 15 cm/s ($t$ test, $P = 0.00008$; $N = 11$) (Fig. 6a). This reduction in speed was characterized by a distinct locomotor phenotype which included reduced peak acceleration ($t$ test, $P = 0.0118$; $N = 11$) (Fig. 6b), decrease in stride length ($t$ test, $P = 0.0036$) (Fig. 6c) and step frequency ($t$ test, $P = 0.0004$) (Fig. 6d) and increased drag on the treadmill (Drag counts $t$ test, $P = 0.0196$; Drag duration $t$ test, $P = 0.0068$) (Fig. 6e, f) compared to controls. Characteristically, the locomotor phenotype was not followed by alterations in the left-right hindlimb coordination. Thus, the left-right hindlimb coordination showed out of phase activity with average phase

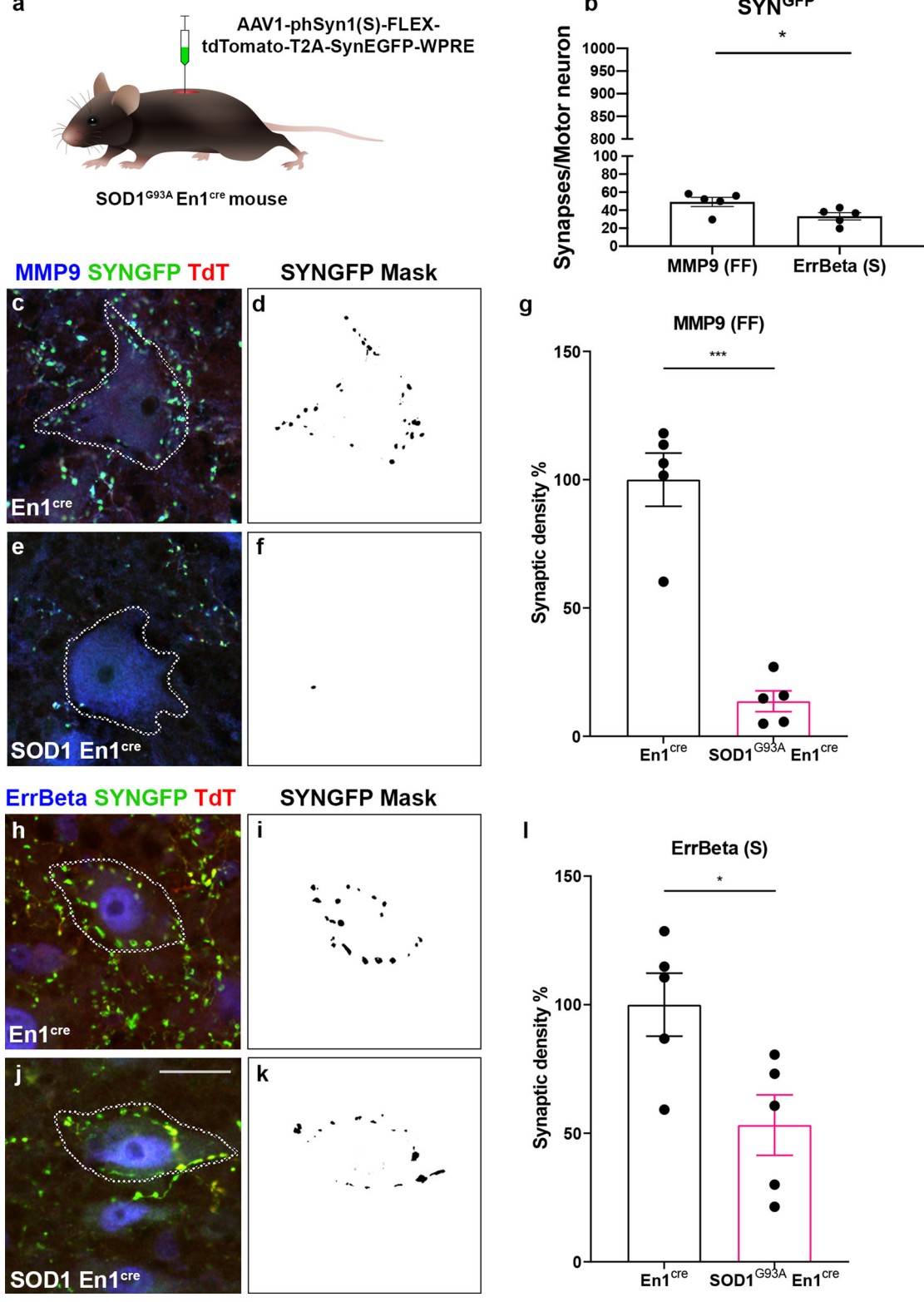

values close to 180 degrees before and after Onset of locomotor phenotype in individual mice (Fig. 5g–i) and for the population of mice (Watson–Williams test, pre-symptomatic vs onset $P = 0.1582$; $n = 15$ independent steps per mouse) (Fig. 6g) similar to wild-type mice (Watson–Williams test, wt vs SOD1[G93A] pre-symptomatic $P = 0.1004$; wt vs SOD1[G93A] onset $P = 0.3261$; $n = 15$ steps) (Fig. 6g). We also analyzed the interlimb coordination

between hindlimbs and forelimbs on the same side (homolateral coordination: left hindlimb versus left forelimb or right hindlimb versus right forelimb) or diagonally (left hindlimb-right forelimb or right hindlimb-left forelimb). While homolateral coordination generally showed out of phase values around 180 degrees there was a significant but small shift in the direction of the coordination between the pre-symptomatic and onset phenotype

**Fig. 3 Fast motor neurons receive stronger V1 interneuron innervation than slow motor neurons and V1 synapses are lost in the SOD1$^{G93A}$ mouse model. a** Experimental approach: an AAV-phSyn1(S)-FLEX-tdTomato-T2A-SypEGFP-WPRE virus was delivered by intraspinal injections in En1$^{cre}$ or SOD1$^{G93A}$;En1$^{cre}$ mice at P42. **b** In En1$^{cre}$ mice, MMP9$^+$ fast motor neurons were found to receive more inputs than ErrBeta$^+$ slow motor neurons (MMP9 = 49.2 ± 11.4 and ErrBeta = 33.2 ± 9; two-tailed $t$ test, $P$ = 0.0401, $t$ = 2.447, df = 8; $N$ = 5 independent mice per condition). Synaptic terminals are GFP$^+$ while En1$^{cre}$ interneurons are TdTomato$^+$ (**c**, **e**, **h**, **j**). Synaptic density analysis performed on fast motor neurons in En1$^{cre}$ (**c**) and SOD1$^{G93A}$;En1$^{cre}$ mice (**e**); masks are shown in (**d**) and (**f**), respectively. Percentage of synaptic density (**g**) in MMP9$^+$ motor neurons revealed a dramatic reduction in GFP$^+$ terminals in SOD1$^{G93A}$ compared to control conditions (two-tailed $t$ test, $P$= 0.00005, $t$ = 7.783, df = 8; $N$ = 5 independent mice per condition). ErrBeta$^+$ motor neurons were analyzed in En1$^{cre}$ (**h**) and SOD1$^{G93A}$;En1$^{cre}$ mice (**j**), and synaptic density was reconstructed (**i**, **k**) and quantified (**l**). Percentage of synaptic density in slow motor neurons showed also a reduction in GFP$^+$ terminals in ALS mice (two-tailed $t$ test, $P$ = 0.0245, $t$ = 2.764, df = 8; $N$ = 5 independent mice per condition). TdTomato in red, GFP in green and MMP9 (**c**, **e**) or ErrBeta (**h**, **j**) in blue. Scale bar in (**j**) =50 μm representative for all images. A minimum of 300 motor neurons was quantified per condition. In all graphs, data are presented as mean values ± SEM. Source data are provided as a Source Data file.

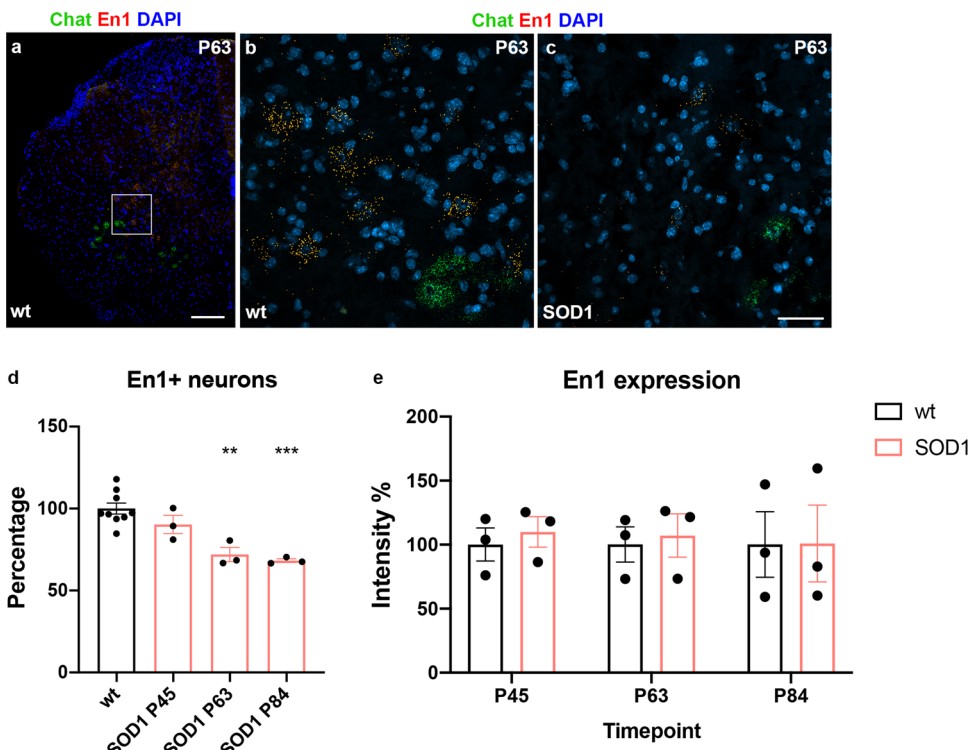

**Fig. 4 En1 positive neuron reduction from postnatal day 63 in the SOD1$^{G93A}$ mouse. a** Microphotograph of a wild-type mouse hemicord after RNAscope in situ hybridization at postnatal day 63. Chat in green, En1 in orange and DAPI in blue. **b** Magnification of Chat and En1 expression in control conditions and **c** at P63 in a SOD1$^{G93A}$ mouse. **d** Quantifications of En1 positive neurons in control conditions versus postnatal day 45, 63, and 84 in the ALS mouse model. There was a reduction at P63 to about 75% and to about 68% at P84 compared to age-matching wild-types (one-way ANOVA and Dunnett's post hoc, $F$ (3, 14) = 13.45, P45 $P$ = 0.3108, P63 $P$ = 0.0010, P84 $P$ = 0.0003; wt $N$ =9, SOD1$^{G93A}$ $N$ = 3 mice per timepoint). **e** Intensity-measurements of En1 positive dots present within the quantified neurons showed no differences among the different timepoints, indicating that En1 neurons still present in the SOD1$^{G93A}$ mouse do not show lower transcript expression at these than earlier timepoints (two-way ANOVA and Tukey's multiple comparisons test, $F$(2, 12) = 0.02754, P45 $P$ = 0.9990, P63 $P$ = 0.9998, P84 $P$ = 1; $N$ = 3 independent mice per condition per timepoint). Scale bar in (**a**) =200 μm, scale bar in (**c**) =50 μm representative also for (**b**). A minimum of 60 pictures (as shown in (**b**–**c**)) was quantified per condition. In all graphs, data are presented as mean values ± SEM. Source data are provided as a Source Data file.

(Watson–Williams test, LH-LF $P$ = 0.00009, RH-RF $P$ = 0.0462; $n$ = 15 steps). In contrast there was no change in the out of phase coordination of the diagonal limb coordination between the pre-symptomatic and onset phenotype (Watson–Williams test, LF-RH $P$ = 0.0880, RF-LH $P$ = 0.8006; $n$ = 15 steps) (Supplementary Fig. 6a–d). While the locomotor performance was clearly affected at these early timepoints, low force grip-strength remained unchanged until later timepoints (Fig. 6h). Together these findings show that motor neuron-specific affection of motor neuron synapses has measurable changes on motor performance that may involve high motor force tasks before low motor force tasks are affected.

**The locomotor phenotype in SOD1$^{G93A}$ mice is mimicked by loss of En1 neuron function**. Since these changes in locomotor phenotype match the locomotor phenotype previously reported after loss of V1 interneuron function in the entire nervous system in wild-type mice[19] we next tested if V1 interneuron synapse loss on motor neurons is involved in the observed locomotor phenotype in SOD1$^{G93A}$ mice. For this, we performed experiments with selective dampening of V1 interneurons activity in the spinal cord only. First, we expressed inhibitory (i) DREADDs (hM$_4$D)—which can be activated by clozapine-N-oxide (CNO) leading to hyperpolarization and dampening of cell excitability[49]—in En1$^+$ neurons. To target En1$^+$ neurons in the spinal cord only we used

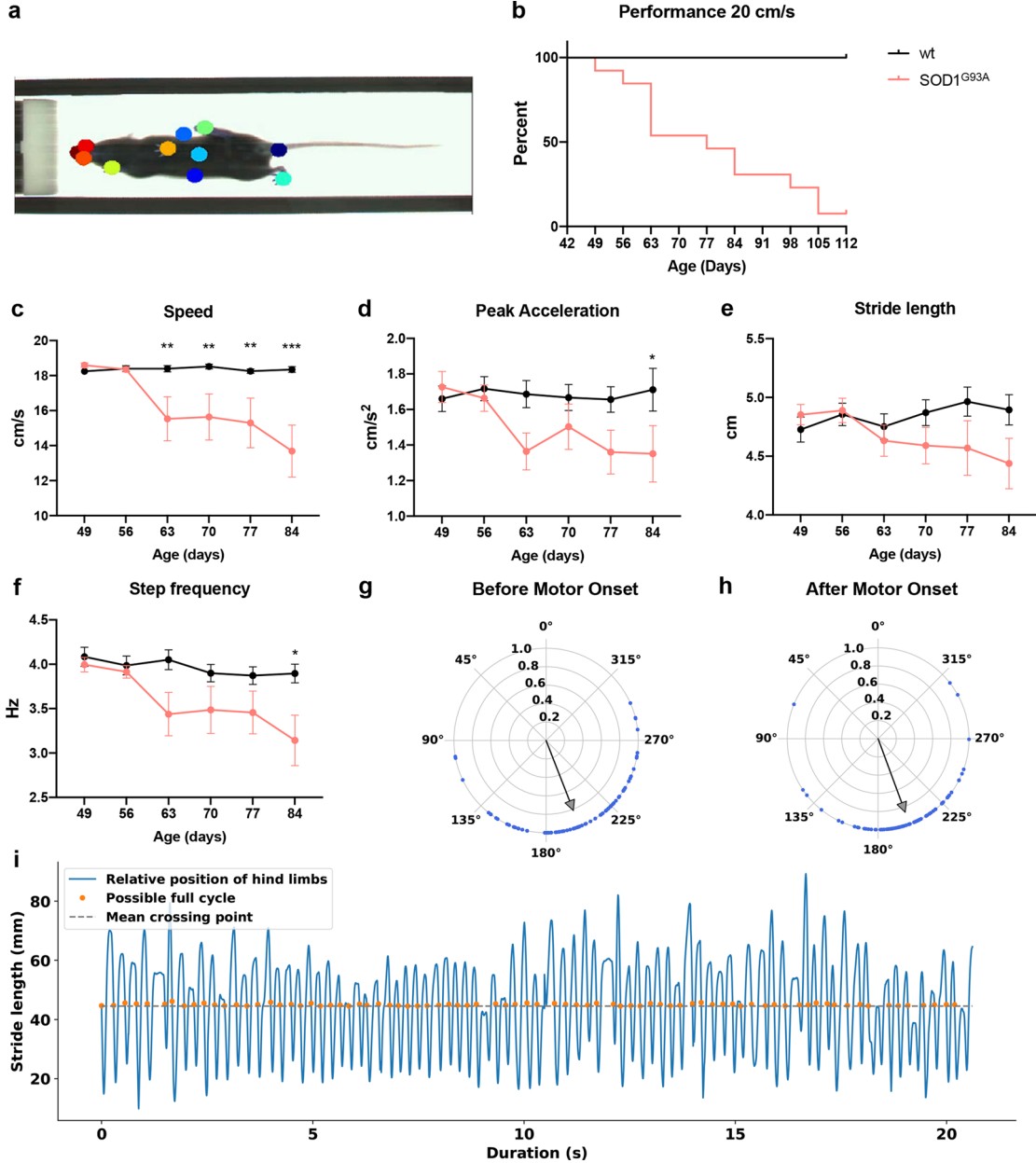

**Fig. 5 SOD1$^{G93A}$ mice between postnatal day 49 and 63 show locomotor deficits. a** Example of tracking approach performed on recorded videos with DeepLabCut analysis tool. Colored circles show digital markers placed on the animals to extrapolate tracks for analysis. **b** Performance of wild-type (wt) and SOD1$^{G93A}$ mice on treadmill at a speed of 20 cm/s. Percentage shows that between P49 and P63 46.2% of the SOD1$^{G93A}$ mice cannot perform the task (median = 77; two-tailed Gehan–Breslow–Wilcoxon test, $P = 0.0003$, Chi square = 12.92, df = 1; $N = 11$ independent mice per condition). At a longitudinal time scale, mice show a progressive reduction of speed (two-way ANOVA and Dunnett's post hoc, $F_{(5, 105)} = 3.020$, P63 $P = 0.0060$, P70 $P = 0.0085$, P77 $P = 0.0027$, P84 $P = 0.00009$; wt $N = 8$ mice, SOD1$^{G93A}$ $N = 15$ mice) compared to control mice (**c**) as well as a progressive reduction in peak acceleration (**d**) (two-way ANOVA and Dunnett's post hoc, $F_{(5, 126)} = 0.9761$, P84 $P = 0.0468$; wt $N = 8$ mice, SOD1$^{G93A}$ $N = 15$ mice), stride length (**e**) (two-way ANOVA and Dunnett's post hoc, $F_{(5, 105)} = 1.441$, $P = 0.2157$; wt $N = 8$ mice, SOD1$^{G93A}$ $N = 15$ mice) and step frequency (**f**) (two-way ANOVA and Dunnett's post hoc, $F_{(5, 105)} = 1.951$, P84 $P = 0.0167$; wt $N = 8$ mice, SOD1$^{G93A}$ $N = 15$ mice) when compared to wild-type mice. Hindlimb left-right alternation was compared before (**g**) and after (**h**) Onset of locomotor phenotype. Blue dots represent single steps of a given animal before and after onset. Gray arrows depict mean vectors of the average direction of all steps. Phase values at 180 degrees correspond to strict alternation. Stride analysis in (**i**) shows the out of phase pattern of the hindlimbs during locomotion which was extracted for coordination analysis. In all graphs, data are presented as mean values ± SEM. Source data are provided as a Source Data file.

a conditional intersectional genetics approach using the R (ROSA26)C(CAG)::FPDi mouse line[49] that allows for dual-recombinase with (F; *Flt*) (P; *LoxP*) that leads to expression of iDREADDs in combinatorially defined neurons. We therefore crossed HoxB8$^{FlipO}$ mice[50]—where HoxB8 is expressed in the spinal cord from cervical level 4 and downwards—with En1$^{cre}$

and R(ROSA26)C(CAG)::FPDi mice to obtain triple En1$^{cre}$; HoxB8$^{FlipO}$;RC::Di mice (Fig. 7a). The successful intersection of *Cre-Lox* and *Flp-Frt* systems resulted in HA-tag expression in the neuronal population of interest (Supplementary Fig. 4a–d). Moreover, we corroborated that the En1 population should be responsive to CNO by showing that En1$^+$ neurons expressed the

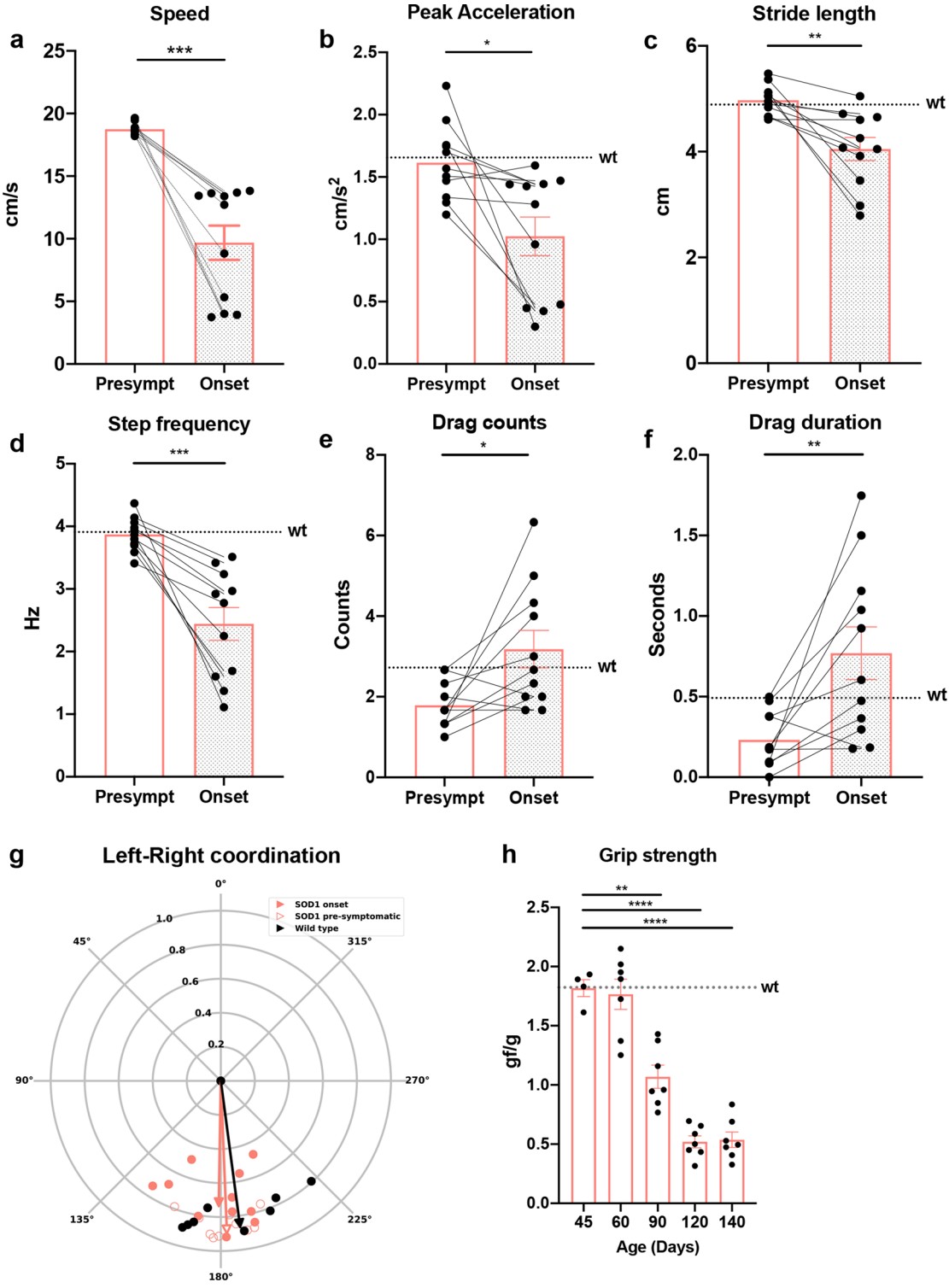

G-protein coupled inwardly rectifying potassium channels (GIRK1 and GIRK2) required for the iDREADD-mediated hyperpolarization[51,52] (Supplementary Fig. 4e, f). CNO was given intraperitoneally (1 mg/kg) as a single dose to triple transgenic mice as well as control mice which were not carrying the *Cre-Lox* and *Flp-Frt* intersectional combination. CNO in this concentration does not have any effect on locomotor performance in wild-type mice[53,54]. Mice were tested before and up to 20 min after administration of CNO on the treadmill with a speed of 20 cm/s. In En1$^{cre}$;HoxB8$^{FlipO}$;RC::Di mice CNO effects were detected between 10 and 15 min after injection (Supplementary

Movie 3 and 4). Seven out of twelve animals responded with a dramatic reduction in speed of locomotion (Fig. 7b), while the other five showed more pronounced changes in step frequency (Fig. 7e; En1$^{cre}$;HoxB8$^{FlipO}$;RC::Di $N = 12$, controls $N = 9$). Interestingly, the phenotype showed by the triple transgenic mice after CNO administration recapitulated the Onset of locomotor phenotype seen in the SOD1$^{G93A}$ mice. The majority of the mice showed reduced speed compared to their performance before injection and 50% of the mice maintained a speed lower than 15 cm/s (one-way ANOVA and Sidak's post hoc, $P = 0.0005$) (Fig. 7b). Moreover, animals showed a decrease in peak

**Fig. 6 Characterization of the Onset of locomotor phenotype.** Performance of SOD1$^{G93A}$ mice before and after Onset of locomotor phenotype is characterized by loss of speed (**a**) (two-tailed $t$ test, $P = 0.00008$, $t = 6.350$, df = 10; $N = 11$ mice), reduction in peak acceleration (**b**) (two-tailed $t$ test, $P = 0.0118$, $t = 3.072$, df = 10; $N = 11$ mice), decrease in stride length (**c**) (two-tailed $t$ test, $P = 0.0036$, $t = 3.779$, df = 10; $N = 11$ mice) and step frequency (**d**) (two-tailed $t$ test, $P = 0.0004$, $t = 5.273$, df = 10; $N = 11$ mice). Moreover, SOD1$^{G93A}$ mice showed increased dragging events when trying to cope with the speed of the belt, drag counts in (**e**) (two-tailed $t$ test, $P = 0.0196$, $t = 2.776$, df = 10; $N = 11$ mice) and drag duration in (**f**) (two-tailed $t$ test, $P = 0.0068$, $t = 3.395$, df = 10; $N = 11$ independent mice). Dotted lines show averages for wild-type (wt) mice in all parameters included in the analysis. **g** Quantifications of left-right alternation showed as circular plots. Perfect alternation corresponds to a phase of 180 degrees. Data from individual animals are plotted for each condition (orange-empty SOD1 pre-symptomatic, orange-full SOD1 onset, black wild-type). The mean vectors for each condition are represented in the respective colors. There is no difference in the mean phase for the different conditions (two-tailed Watson–Williams test, SOD1$^{G93A}$ pre-symptomatic vs SOD1$^{G93A}$ onset $P = 0.1582$; wt vs SOD1$^{G93A}$ pre-symptomatic $P = 0.1004$; wt vs SOD1$^{G93A}$ onset $P = 0.3261$; $N = 11$ mice, $n = 15$ steps per mouse). **h** Grip strength shows progressive but late decline of low force performance in the SOD1$^{G93A}$ starting from P90 (one-way ANOVA and Dunnett's post hoc, $F (4, 27) = 45.89$, P90 $P = 0.00009$, P120 $P = 0.00009$, P140 $P = 0.00009$; P45 $N = 4$, $N = 7$ mice for all other timepoints). In all graphs, data are presented as mean values ± SEM. Source data are provided as a Source Data file.

acceleration (one-way ANOVA and Sidak's post hoc, $P = 0.0064$) (Fig. 7c) reduced stride length (one-way ANOVA and Sidak's post hoc, $P = 0.0355$) (Fig. 7d) and step frequency (one-way ANOVA and Sidak's post hoc, $P = 0.0013$) (Fig. 7e). Mice also increased the duration of the dragging events (one-way ANOVA and Sidak's post hoc, Drag duration $P = 0.0416$) but not the absolute number of dragging events (one-way ANOVA and Sidak's post hoc, Drag counts $P = 0.2578$) (Fig. 7f–g). Similar to wild-type animals En1$^{cre}$;HoxB8$^{FlipO}$;RC::Di left-right hindlimb coordination had phase values around 180 degrees corresponding to alternation and there was no change after CNO administration (Watson–Williams test, $P = 0.1294$; $n = 15$ steps) (Fig. 7i). As for the SOD1$^{G93A}$ mice at Onset of locomotor phenotype, the homolateral coordination differed before and after CNO administration (Watson–Williams test, LH-LF $P = 0.0481$, RH-RF $P = 0.00009$; $n = 15$ steps) although the phase values were still supporting alternation (Supplementary Fig. 6e–f). The diagonal coordination was also less affected by CNO administration as observed in SOD1$^{G93A}$ mice after onset (Watson–Williams test, LF-RH $P = 0.8453$, RF-LH $P = 0.0059$; $n = 15$ steps) (Supplementary Fig. 6g–h). Interestingly, the CNO administration did not change hindlimb grip strength in En1$^{cre}$;HoxB8$^{FlipO}$;RC::Di which was unchanged before and after drug administration and comparable to control animals (one-way ANOVA and Sidak's post hoc, $P = 0.6486$) (Fig. 7h). All together these results show that a dampening of the En1$^+$ neuron activity in the spinal cord alone leads to a reversible slowing of the locomotor speed in intact mice similar to what is seen in SOD1$^{G93A}$ mice and as previously shown when inactivating En1$^+$ neurons in the entire nervous system[19].

**Hyperflexion of the ankle and knee joints is found both in SOD1$^{G93A}$ mice at onset of locomotor phenotype and after silencing of En1$^+$ interneurons.** To further elucidate the similarities between the SOD1$^{G93A}$ at Onset of locomotor phenotype and the En1$^{cre}$;HoxB8$^{FlipO}$;RC::Di mice phenotype after CNO administration, we investigated the angular positions of the hindlimb joints during locomotion. Hyperflexion of the hindlimb joints was previously reported during walking following V1 interneuron ablation[21]. To detect possible angular changes of hip, knee, ankle, and foot joints, kinematic analysis was performed in SOD1$^{G93A}$ mice after Onset of locomotor phenotype and compared to wild-type littermates (Fig. 8a). Stick diagrams of two consecutive step cycles (Fig. 8b-c) show changes in SOD1$^{G93A}$ joint positions revealing hyperflexion during mid-swing and early stance phases. Analysis of the angles maintained by the joints in the stance-swing phases reveals changes in the knee (Fig. 8l–m, p), ankle (Fig. 8r–s, v), and foot (Fig. 8x–y, ab) position, but not in the hip (Fig. 8f–g, j) ($t$ test, hip $P = 0.2059$, knee $P = 0.0196$, ankle $P = 0.0483$, foot $P = 0.0052$; wt $N = 8$, SOD1$^{G93A}$ $N = 5$) as

previously reported when ablating V1 interneurons. Notably, the same changes in the angular position of the hindlimb joints were observed after administration of (1 mg/kg) CNO in En1$^{cre}$;HoxB8$^{FlipO}$;RC::Di mice (Fig. 8d–e). Swing-stance phases angular changes were found in the knee (Fig. 8n–o, q), ankle (Fig. 8t–u, w), and foot (Fig. 8z–aa, ac) joints but not in the hip (Fig. 8h–i, k) ($t$ test, hip $P = 0.7901$, knee $P = 0.0212$, ankle $P = 0.0140$, foot $P = 0.0071$; $N = 7$ for both conditions). Therefore the kinematic analysis of both SOD1$^{G93A}$ after Onset of locomotor phenotype and En1$^{cre}$;HoxB8$^{FlipO}$;RC::Di mice after CNO administration showed clear changes during swing phase with hyperflexion in limb joints that corresponds to the affection of the V1 population during locomotion[21].

**Dampening of En1$^+$ interneuron activity in SOD1$^{G93A}$ mice after Onset of locomotor phenotype does not change the locomotor phenotype.** We next set out to evaluate the contribution of V1 neurons to the locomotor phenotype in SOD1$^{G93A}$ mice after their Onset of locomotor phenotype. We hypothesized that if the loss of V1 neuron connectivity is the main course of the locomotor phenotype observed in SOD1$^{G93A}$ mice no further worsening of the phenotype should be observed by dampening V1 activity after Onset of locomotor phenotype. To test this we crossed En1$^{cre}$;HoxB8$^{FlipO}$;RC::FPDi mice with SOD1$^{G93A}$ mice (Fig. 9a). Out of twelve SOD1$^{G93A}$ positive mice four carried both the *Cre-Lox* and *Flp-Frt* intersectional combination and the remaining littermates were used as control. The SOD1$^{G93A}$;En1$^{cre}$;HoxB8$^{FlipO}$;RC::FPDi showed the same life span as the pure SOD1$^{G93A}$ animals (Supplementary Fig. 1c; 155.5 ± 10.7 days). The SOD1$^{G93A}$;En1$^{cre}$;HoxB8$^{FlipO}$;RC::FPDi and littermate controls were tested weekly on a treadmill from P42 until P98 (Supplementary Fig. 5a–e) (median = 77; Gehan-Breslow-Wilcoxon test $P = 0.00002$; control $N = 9$, SOD1$^{G93A}$; En1$^{cre}$;HoxB8$^{FlipO}$;RC::FPDi $N = 11$) (Supplementary Movie 5). During this time all SOD1$^{G93A}$ animals showed Onset of locomotor phenotype (Supplementary Movie 6) which also in this case was characterized by a decrease in speed ($t$ test, $P = 0.00009$; $N = 11$), reduced peak acceleration ($t$ test, $P = 0.0060$; $N = 11$), decreased stride length ($t$ test, $P = 0.0002$), decreased step frequency ($t$ test, $P = 0.0020$) as well as an increased number of dragging events ($t$ test, Drag counts $P = 0.0034$, Drag duration $P = 0.0195$), and remaining left-right hindlimb alternation not different from controls (Watson–Williams test, pre-symptomatic vs onset $P = 0.2792$; $n = 15$ steps per mouse) (Supplementary Fig. 5f–l). Moreover, changes in the homolateral coordination and the lack of changes in the diagonal limb coordination were comparable to the one observed in SOD1$^{G93A}$ mice at onset (Watson–Williams test, homolateral LH-LF $P = 0.0097$, RH-RF $P = 0.0216$, diagonal RF-LH $P = 0.2389$, LF-RH $P = 0.2376$; all $n = 15$ steps) (Supplementary Fig. 5m–p). On the day the animals

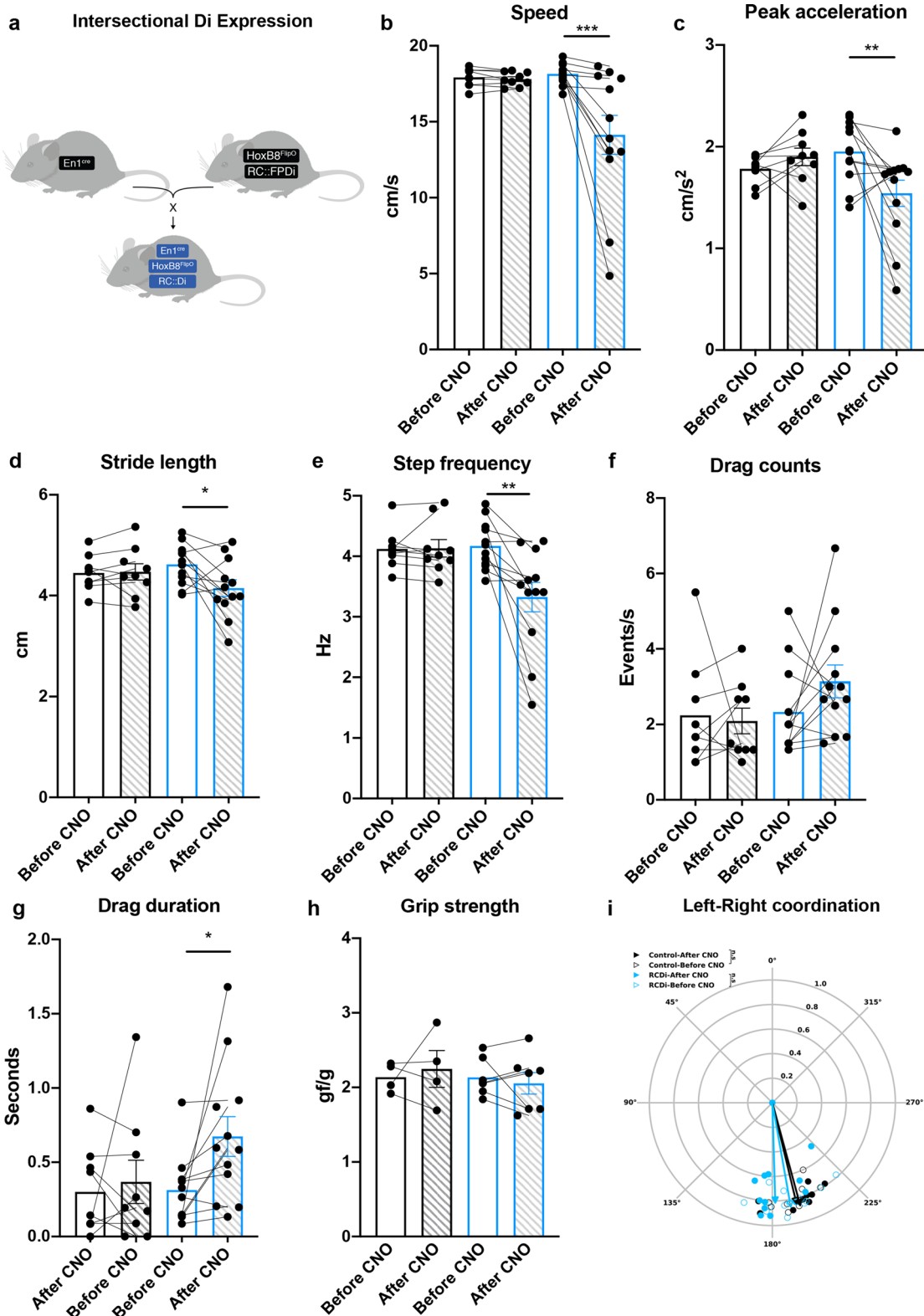

reached the Onset of locomotor phenotype, 1 mg/kg CNO was administrated as a single dose and mice were tested 10–15 min after injection at a speed suitable for their phenotype. As expected, the SOD1$^{G93A}$ mice without active iDREADD receptor showed no change in any locomotor parameters after CNO (Fig. 9b–i, orange bars). Similarly, the locomotor phenotype of the SOD1$^{G93A}$;En1$^{cre}$;HoxB8$^{FlipO}$;RC::Di mice did not differ before and after CNO administration (Supplementary Movie 7),

since speed (Fig. 9b, magenta bars), stride length (Fig. 9d, magenta bars) and step frequency (Fig. 9e, magenta bars) remained unchanged. Moreover, also peak acceleration (Fig. 9c, magenta bars), drag events (Fig. 9f–g magenta bars), left-right hindlimb coordination (Watson–Williams test, $P = 0.3682$; $n = 15$ steps) (Fig. 9i, magenta dots), homolateral limb coordination (Watson–Williams test, LH-LF $P = 0.5051$, RH-RF $P = 0.1575$; $n = 15$ steps) (Supplementary Fig. 6i–j) and the diagonal

**Fig. 7 Dampening of spinal V1 interneuron activity recapitulates the Onset of locomotor phenotype. a** Mouse genetic approach used to express inhibitory DREADDs specifically in En1[+] spinal interneurons. En1[cre] mouse strain was crossed with HoxB8[FlipO];RC::FPDi animals. After CNO administration, En1[cre];HoxB8[FlipO];RC::Di mice (blue bars) show loss of locomotor speed **b** compared to their performance before administration of CNO (one-way ANOVA and Sidak's post hoc, $F(3, 38) = 7.058$, $P = 0.0005$; controls $N = 9$ mice, En1[cre];HoxB8[FlipO];RC::Di $N = 12$ mice). Dual conditional mice also showed decrease of peak acceleration (**c**) (one-way ANOVA and Sidak's post hoc, $F(3, 38) = 3.792$, $P = 0.0064$; controls $N = 9$ mice, En1[cre];HoxB8[FlipO];RC::Di $N = 12$ mice) and reduction in stride length (**d**) (one-way ANOVA and Sidak's post hoc, $F(3, 38) = 2.137$, $P = 0.0355$; controls $N = 9$ mice, En1[cre];HoxB8[FlipO]; RC::Di $N = 12$ mice) and step frequency (**e**) (one-way ANOVA and Sidak's post hoc, $F(3, 38) = 6.148$, $P = 0.0013$; controls $N = 9$ mice, En1[cre];HoxB8[FlipO]; RC::Di $N = 12$ mice). Mice also showed increased dragging duration (one-way ANOVA and Sidak's post hoc, $F(3, 38) = 2.625$, $P = 0.0416$; controls $N = 9$ mice, En1[cre];HoxB8[FlipO];RC::Di $N = 12$ mice) (**g**), although they did not show an increased number of drags (**f**) (one-way ANOVA and Sidak's post hoc, $F(3, 38) = 1.436$, $P = 0.2578$; controls $N = 9$ mice, En1[cre];HoxB8[FlipO];RC::Di $N = 12$ mice). Grip strength remained unchanged before and after CNO administration (**h**) (one-way ANOVA and Sidak's post hoc, $F(3, 18) = 0.2763$, $P = 0.6486$; control $N = 4$, En1[cre];HoxB8[FlipO];RC::Di $N = 7$ mice) as well as left-right alternation (blue-empty = En1[cre];HoxB8[FlipO];RC::Di before CNO; blue-full = En1[cre];HoxB8[FlipO];RC::Di after CNO; black-empty = control before CNO; black-full = control after CNO) (**i**) (two-tailed Watson–Williams test, $P = 0.1294$; controls $N = 9$ mice, En1[cre];HoxB8[FlipO];RC::Di $N = 12$ mice, $n = 15$ steps per mouse). CNO administration did not have any effect on control animals (black bars and black arrows; $N = 9$ mice). In all graphs, data are presented as mean values ± SEM. Source data are provided as a Source Data file.

coordination (Watson–Williams test, RF-LH $P = 0.1608$, LF-RH $P = 0.2757$; $n = 15$) (Supplementary Fig. 6k–l) remained unaffected by CNO administration. CNO administration did not have any effects on grip strength (Fig. 9h). All together, these results show that the slowing of locomotion caused by V1 neuron inactivation by inhibitory iDREADDs is occluded in the SOD1[G93A] animals after Onset of locomotor phenotype suggesting that the loss of En1[+] terminals on fast motor neurons is the direct cause of the slowing of locomotor phenotype in SOD1[G93A] mice.

## Discussion

The present study uncovers an asymmetric innervation pattern by inhibitory glycinergic spinal interneurons of fast and slow motor neurons. This preferential innervation pattern of fast versus slow motor neurons was selectively lost during early ALS disease stages both for the general population of inhibitory spinal neurons and for the large population of inhibitory V1 neurons. This motor neuron-specific synapse loss mirrors the preferential vulnerability of fast motor neurons in ALS when compared to the slow motor neurons. Remarkably, the loss of inhibitory synaptic inputs is paralleled by a distinct locomotor phenotype in SOD1[G93A] mice which is phenocopied by the reversible dampening of V1 interneuron activity in wild-type mice, but it is occluded in SOD1[G93A] mice after onset of the locomotor phenotype. These findings reveal a distinct locomotor phenotype in ALS that may be linked to V1 neuron synapse retraction on fast motor neurons during the early stages of the disease. The retraction appears before motor neuron death and changes in the neuromuscular junctions, suggesting that, our work, has additionally uncovered a potential source of the difference in the vulnerability of fast and slow motor neurons within the spinal cord in ALS.

Several previous synaptic density investigations performed by immunohistochemistry in SOD1[G93A] mice have shown a reduction in glycinergic but not Gabaergic inputs onto α-motor neurons starting from postnatal day 60[9–13]. None of these studies have, however, differentiated between inputs to fast and slow motor neurons and has therefore been unable to reveal the twofolds higher innervation of fast versus slow, and the selective loss of inhibitory synapses onto fast motor neurons. The observed changes were not due to the differential size of fast and slow motor neurons because they were also present when the synaptic loss was normalized to the motor neuron area. The motor neuron-specific loss is similar for the inhibitory last order V1-population of interneurons that is composed of 80% glycinergic neurons and that make up more than 50% of soma-near motor neuron synapses[20,44]. These findings suggest that V1 population of neurons make up a substantial part of the synapse retraction

although further studies are needed to reveal if alterations in connectivity are restricted to the glycinergic V1 interneurons. However, we did not observe alterations in left-right alternation or diagonal interlimb coordination after the Onset of locomotor phenotype in SOD1[G93A] mice, thus, we can assume that glycinergic commissural V0 interneurons are not affected at this timepoint. We did find a significant drop in En1 positive neurons starting from postnatal day 63 indicating that there is an early degeneration of the V1 subpopulation in SOD1[G93A] that outnumber the reduction of V1 neurons during normal aging. However, a large proportion of En1[+] neurons are still present in aging mice and that proportion do not show further changes in transcript expression at later (>P63) disease stages. The neuron loss is percentage-wise less than the synaptic loss on motor neurons which suggests that there is a retraction of synapses from motor neurons. Alternatively, the V1 premotor inputs to slow and fast motor neurons belong to different subpopulations of V1 neurons[26] and it is the V1 neurons to fast motor neurons that are dying. At the moment we have no way to test this directly.

We did not find any major change in glutamatergic input to motor neurons during the progression of the disease. Previous studies have also concluded that the cholinergic innervation of motor neurons remain unchanged in SOD1[G93A] mice during disease development. However, the pre-symptomatic loss of soma-near inhibitory synapses in itself is likely to reduce the inhibitory/excitatory synapse balance on fast motor neurons compared to slow motor neurons which may lead to intrinsic motor neuron increased excitability observed in pre-symptomatic SOD1[G93A] mice[55] and higher stress responses in vulnerable fast motor neurons in ALS[3,5,56,57]. Moreover, the recent finding that loss of the En1 transcription factor expressed in V1 neurons may lead to motor neuron degeneration[58] may be another mechanism that contributes to the ALS disease progression when the inhibitory synapses are retracted. The actual cause of the retraction is not known but the study underscores that ALS may start as an interneuron affection.

A profound finding of this study is that the SOD1[G93A] mice exhibited a locomotor phenotype characterized by loss of speed of locomotion and reduction in stride length. Moreover, the segmental left-right alternation and diagonal hindlimb and forelimb alternation was unchanged in SOD1[G93A] mice, while there were small changes in homolateral coordination. In contrast the intralimb coordination showed hyperflexion in most joints. This locomotor phenotype was observed before motor neuron death and NMJ loss and before loss of grip strength which suggests that the locomotor phenotype is a sensitive indicator of early ALS progression. Some of these locomotor symptoms have previously been observed in ALS patients with both spinal and bulbar

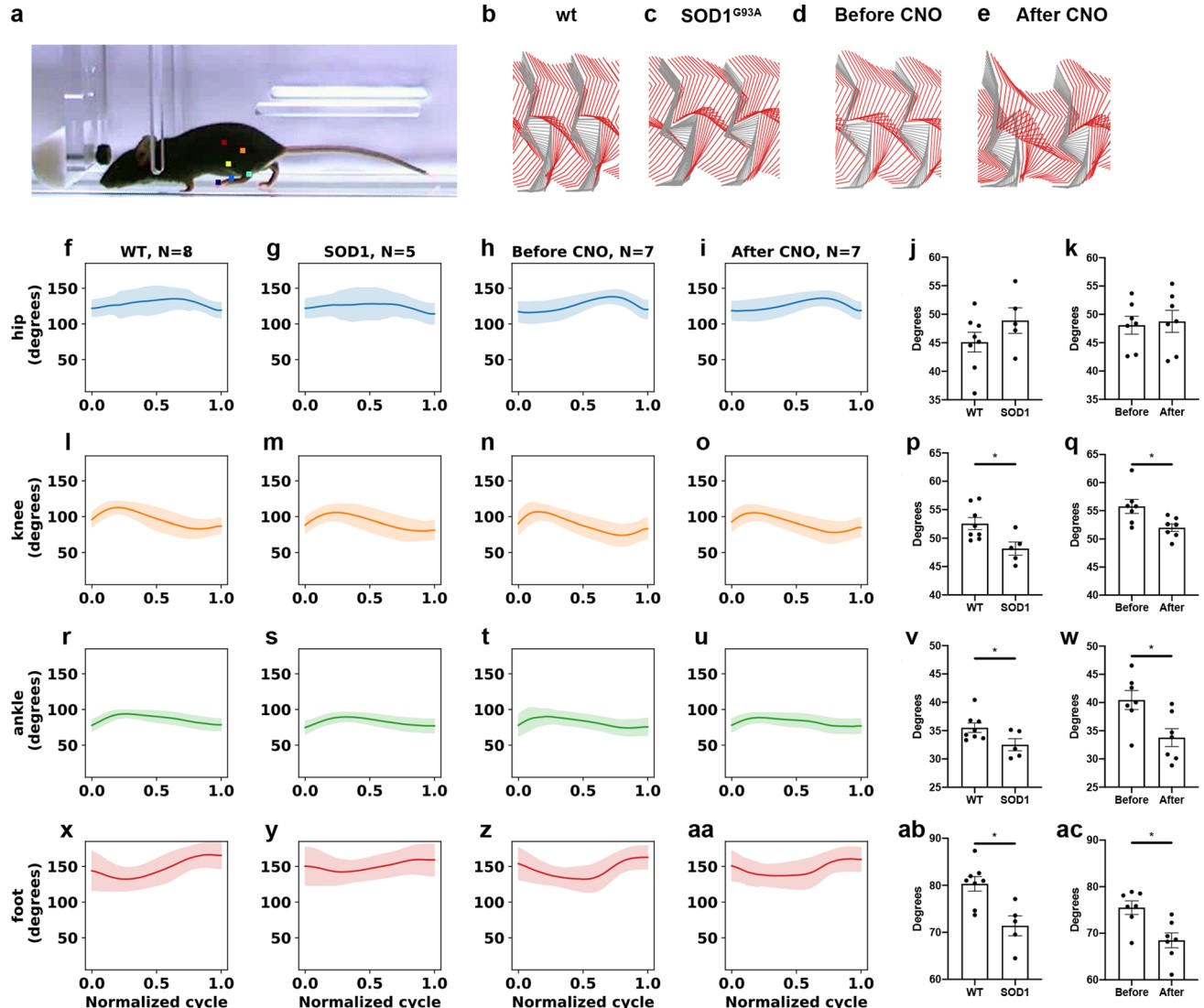

**Fig. 8 Ankle and knee joints show hyperflexion in SOD1[G93A] mice and after silencing of En1 positive neurons. a** Photograph showing the lateral view of a mouse analyzed with DeepLabCut tracking software. Dots mark iliac crest, hip, knee, ankle, foot and toe. Stick figures depicting the relative position of the analyzed joints in wild-type (**b**) and in SOD1[G93A] mice after Onset of locomotor phenotype (**c**) during a complete step cycle, stance phase in gray. Stick figures of En1[cre];HoxB8[FlipO]; RC::Di mice before (**d**) and after (**e**) CNO treatment show similar changes in hyperflexion of the ankle and knee joints as in the SOD1[G93A] mice after onset. Quantification of the changes in angles of hip in wild-type versus SOD1[G93A] conditions (**f–g**, **j**) show no differences (two-tailed $t$ test, $P = 0.2059$, $t = 1.344$, df = 11), while knee (**l–m**, **p**) (two-tailed $t$ test, $P = 0.0196$, $t = 2.729$, df = 11), ankle (**r–s**, **v**) (two-tailed $t$ test, $P = 0.0483$, $t = 2.221$, df = 11) and foot (**x–y**; **ab**) (two-tailed $t$ test, $P = 0.0052$, $t = 3.469$, df = 11) are significantly different between the two conditions (wt $N = 8$ mice, SOD1[G93A] $N = 5$ mice). Quantifications of the joint angles after CNO administration mimics the SOD1[G93A] mice after onset. While no changes were observed in the hip angle (**h–i**, **k**) (two-tailed $t$ test, $P = 0.7901$, $t = 0.2722$, df = 12) angles of knee (**n–o**, **q**) (two-tailed $t$ test, $P = 0.0212$, $t = 2.650$, df = 12), ankle (**t–u**; **w**) (two-tailed $t$ test, $P = 0.0140$, $t = 2.873$, df = 12) and foot (**z–aa**; **ac**) (two-tailed $t$ test, $P = 0.0071$, $t = 3.239$, df = 12) were significantly different before and after CNO treatment (En1[cre];HoxB8[FlipO];RC::Di $N = 7$ independent mice). In all graphs, data are presented as mean values ± SEM. Error bands in (**f–i**), (**l–o**), (**r–u**) and (**s–aa**) represent SD. Source data are provided as a Source Data file.

onsets[59]. Notably, the double conditionals En1[re];HoxB8[FlipO];RC::Di mice after CNO administration had a very similar locomotor phenotype on these measured parameters, which suggest that these symptoms are specific for V1 interneurons depletion. Also, the higher dragging events could be detected both in SOD1[G93A] animals after Onset of locomotor phenotype and in the double conditional mice after CNO administration. Finally, the occlusion of the V1 locomotor phenotype after Onset of locomotor phenotype in ALS strongly suggest that the V1 loss of synapses onto fast motor neurons is the direct cause of the locomotor phenotype observed in ALS.

Loss of motor units in *tibialis anterior* and medial *gastrocnemius* has been reported from postnatal day 40[4] and decreased rotarod performance was found from P60[60]. These symptoms have been generally related to changes in muscle innervation, however we found that the V1 synapses withdraw from fast motor neurons before motor neuron degeneration and also before NMJ denervation. Since divergent results were reported on the timepoint of significant NMJ denervation in fast-twitch fatigable muscles by studies utilizing either different (high expressors)[5,61] or unspecified[40] SOD1[G93A] mouse models, we analyzed NMJ denervation in *tibialis anterior*, *gastrocnemius* and *soleus* muscles

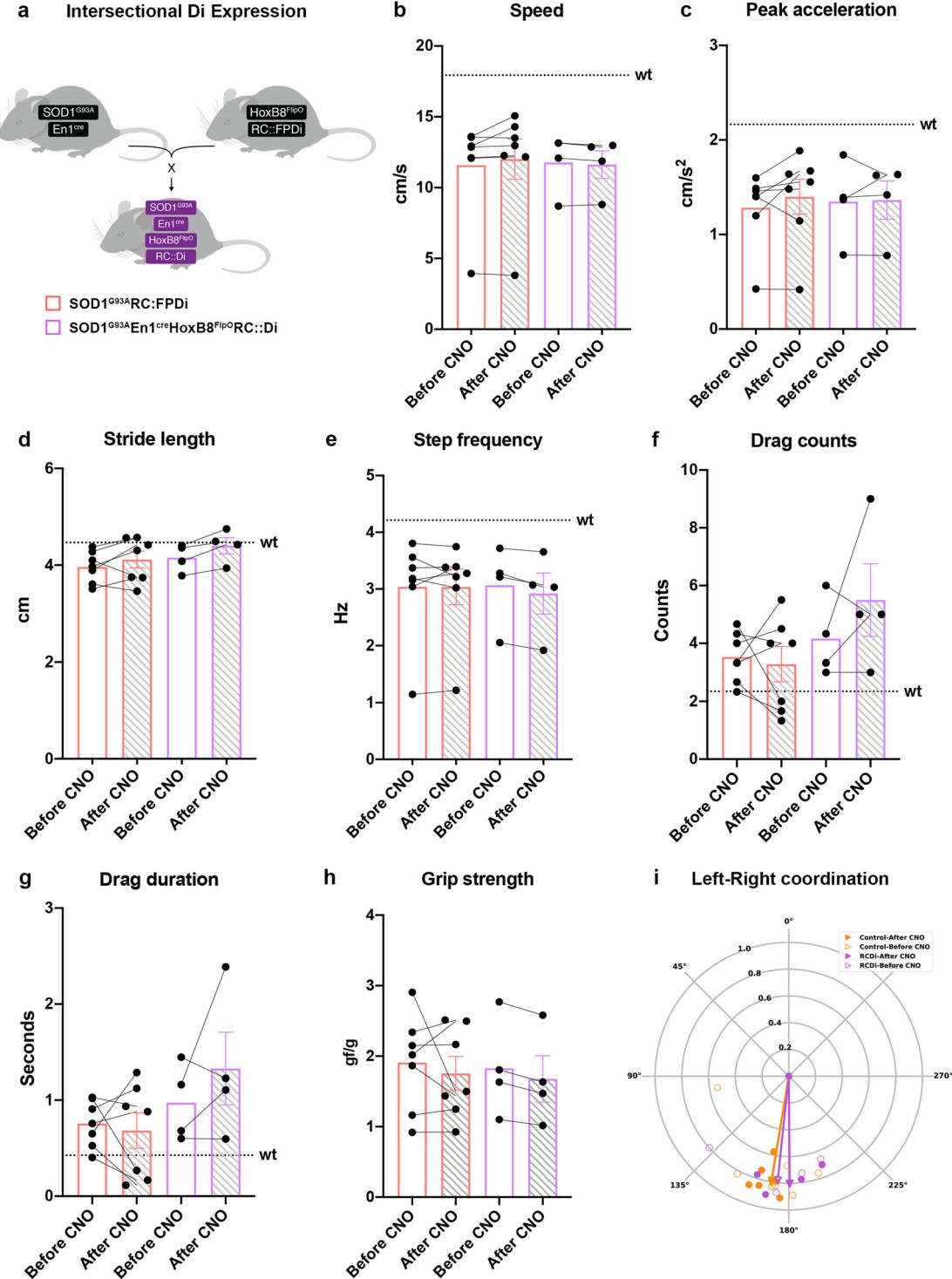

in our SOD1$^{G93A}$ and SOD1$^{G93A}$;GlyT2$^{GFP}$ mice kept on C57Bl6/J congenic background. This analysis demonstrated that glycinergic synapses onto fast motor neurons are lost before NMJ denervation suggesting that the locomotor phenotype we report is not due to retraction of NMJs but linked to the interneuron changes. Moreover, the grip strength analysis replicated results previously obtained when testing low task forces (significant differences from P90)[60].

Since the loss of connectivity between V1 inhibitory interneurons and fast motor neurons appears during early ALS disease progression in SOD1$^{G93A}$ mice and leads to locomotor deficits,

we suggest that future endeavors could focus on rescuing the synaptic connectivity by stabilizing the V1 soma-near synapses. Thus, stabilization of V1 inhibitory inputs could potentially reduce the stress responses in motor neurons. Several attempts have been directed to stabilize the neuromuscular synapses formed by motor neurons and their target muscles[62–65]. Drugs known to reduce hyperexcitability directly in motor neurons as the FDA-approved Riluzole and Retigabine[66] have been shown to improve motor neuron survival in ALS. Although, if the motor neuron hyperexcitability is also induced by the loss of inhibitory synaptic inputs, it would be beneficial to attempt preventing the

**Fig. 9 Dampening of spinal V1 interneuron activity does not have an effect in SOD1$^{G93A}$ mice after Onset of locomotor phenotype. a** Mouse crossing utilized to assess specific spinal V1 interneuron silencing in a SOD1$^{G93A}$ mouse model. SOD1$^{G93A}$;En1$^{cre}$ mice were crossed with HoxB8$^{FlipO}$;RC::FPDi. SOD1$^{G93A}$ mice that did not carry the intersectional expression were used as controls. After CNO administration quadruple transgenics (magenta bars) did not show changes in speed (**b**) (one-way ANOVA and Sidak's post hoc, $F(3, 18) = 0.02417$, $P = 0.9977$; control $N = 7$ independent mice, quadruple transgenics $N = 4$ independent mice) nor in peak acceleration (**c**) (one-way ANOVA and Sidak's post hoc, $F(3, 18) = 0.08636$, $P = 0.9982$; control $N = 7$ mice, quadruple transgenics $N = 4$ mice). Also stride length (**d**) (one-way ANOVA and Sidak's post hoc, $F(3, 18) = 1.211$, $P = 0.5915$; control $N = 7$ mice, quadruple transgenics $N = 4$ mice) and step frequency (**e**) remained unchanged (one-way ANOVA and Sidak's post hoc, $F(3, 18) = 0.02689$, $P = 0.9602$; control $N = 7$ mice, quadruple transgenics $N = 4$ mice). SOD1$^{G93A}$;En1$^{cre}$;HoxB8$^{FlipO}$;RC::Di mice did not show more dragging events (**f–g**) after CNO administration (drag counts one-way ANOVA and Sidak's post hoc, $F(3, 18) = 1.914$, $P = 0.4302$; drag duration one-way ANOVA and Sidak's post hoc, $F(3, 18) = 1.839$, $P = 0.5068$; control $N = 7$ mice, quadruple transgenics $N = 4$ mice). **h** CNO administration did not have any effects on grip strength in SOD1$^{G93A}$;En1$^{cre}$;HoxB8$^{FlipO}$;RC::Di mice (one-way ANOVA and Sidak's post hoc, $F(3, 18) = 0.1216$, $P = 0.9380$; control $N = 7$ mice, quadruple transgenics $N = 4$ mice). SOD1$^{G93A}$ not carrying dual intersectional expression did not show any changes after CNO administration (orange bars). Dotted lines show averages for wild-type (wt) animals in all parameters included in the analysis. **i** Left-right alternation remained unaltered after CNO administration in both SOD1$^{G93A}$;En1$^{cre}$;HoxB8$^{FlipO}$;RC::Di mice and SOD1$^{G93A}$ controls (two-tailed Watson–Williams test, $P = 0.3682$; $n = 15$ independent steps per mouse). In all graphs, data are presented as mean values ± SEM. Source data are provided as a Source Data file.

loss of synaptic inputs early during ALS progression, e.g., by overexpressing either genes involved in synaptogenesis or in vesicular trafficking and exocytosis in inhibitory interneurons.

## Methods

**Ethical permits and mouse strains**. All experiments were in accordance with the EU Directive 20110/63/EU and approved by the Danish Animal Inspectorate - Dyreforsøgstilsynet (Ethical permit: 2018-15-0201-01426) and the local ethics committee at the Faculty of Health and Medical Sciences. SOD1$^{G93A}$ (B6.Cg-Tg (SOD1-G93A)1Gur/J) were retrieved from Jackson Laboratory stock no: #004435 and genotyped following the supplier indications, including copy number quantification of the human mutated SOD1 by qPCR. For qPCRs, the positive control was the SOD1$^{G93A}$ founder breeder obtained from JAX carrying 25 copies of the human mutated SOD1 gene, while a SOD1$^{127X}$ strain carrying 19 copies of the human mutated SOD1 gene[67] was used as a negative control. All mouse strains were bred with congenic C57BL6/J mice, stock no: #000664 (Jackson Laboratory). For multiple crossing the following mice kept on C57BL6/J background were used: GlyT2$^{GFP}$ (obtained from Hanns Zeilhofer's laboratory)[27], HoxB8$^{FlipO}$ (generated in the Kiehn laboratory)[50], R26R-EYFP stock number: #006148 (Jackson Laboratory). RC::FPDi mice[49] was a kind gift from Prof. Susan M Dymecki, Harvard Medical School. The En1$^{cre}$ transgenic mice[25], kept on a C57BL6/J background, were generously obtained from Assistant Prof. Jay Bikoff (St. Jude Children's hospital, St Louis, Texas, USA). Besides genotype and hSOD1 copy number, SOD1 phenotype, including weight loss and survival, was also analysed for all the different crossing. For survival analysis, the humane endpoint was defined as a weight loss of 15% and/or functional paralysis in both hindlimbs together with inability to perform a righting test in <20 s[68]. Mice were housed according to standard conditions: fed ad libitum, constant access to water and a 12:12 h light/dark cycle, at a temperature between 23–24 °C and 45–65% humidity. Both males and females were included in the study and non-transgenic SOD1$^{G93A}$ littermates were used as controls.

**Intraspinal injections and viral delivery**. For viral transfection of the En1-expressing spinal interneurons of the spinal cord, P42 mice were anaesthetized with 2% isoflurane and the lumbar level of the spinal cord was exposed for stereotaxic injections (Neurostar). A small incision was performed with micro-scissors between the T12 and the T13 vertebrae in order to deliver the virus to the lumbar spinal cord. For visualization, the virus was mixed with 4% fast green (Invitrogen) dissolved in saline and injected using a glass micropipette at a rate of 100 nl/min. The micropipette was kept in place for 2 min after viral delivery to avoid backflow. Bilateral injections of an AAV1-phSyn1(S)-FLEX-tdTomato-T2A-SypEGFP-WPRE ($5.56 \times 10^{11}$ /ml, Viral Vector Core, Salk Institute for Biological Sciences; Addgene 51509) were performed in SOD1$^{G93A}$;En1$^{cre}$ mice and wt;En1$^{cre}$ littermates to assess synaptic connectivity of En1 positive neurons to fast fatigable and slow motor neurons. Mice were treated post-operatively with Buprenorphine (Tamgesic) diluted in saline at a concentration of 0.3 mg/ml for pain relief. Three weeks after injections at postnatal day P63 mice were perfused and the tissue was processed for immunohistochemistry (see spinal cord immunohistochemistry).

**Spinal cord immunohistochemistry**. Mice were anesthetized with an overdose of Pentobarbital (250 mg/kg) and perfused transcardially with pre-chilled phosphate-buffered saline (PBS, Gibco) followed by pre-chilled 4% paraformaldehyde (PFA, HistoLab). Spinal cords were dissected and cryoprotected in 30% sucrose dissolved in PBS. Tissue was then sectioned at 30 μm thickness on a cryostat (Thermo Fisher Scientific). Coronal sections were collected on Superfrost Plus slides (Thermo Fisher Scientific) and then stained by immunohistochemistry. Slides were washed for 10 min in PBS at room temperature and blocked for 1 h in PBS + 0.1% Triton-

X100 (PBS-T, Sigma Aldrich) and 1.5% donkey or goat serum (AB_2337258, AB_2336990, Jackson Laboratory). When utilizing monoclonal mouse antibodies, blocking was performed over night at 4 °C with Fab Fragment anti-mouse (115-007-003, Jackson Laboratory) at a concentration 1:50 in PBS-T and 1.5% donkey serum to reduce unspecific background. Sections were then incubated for 48 h at 4 °C in primary antibodies diluted in blocking solution (Supplementary Table 1). Slides were washed three times in PBS at room temperature for 10 min each and incubated with appropriate secondary antibodies (1:500, Alexa Fluor 488, 568, 647 – A3272, A3273, A32931, A32849, A1107 - Invitrogen) diluted in blocking solution for 1 h. Counterstaining was performed either with Hoechst 33342 (1:2000, H3570, Invitrogen) or NeuroTrace 435 (1:200, N21479, Invitrogen). For synaptic density counterstaining, a Synaptophysin antibody Alexa Fluor 594 conjugated was incubated overnight (1:100, sc-17750 Santa Cruz). After three washes is PBS, slides were dried overnight, and cover slipped using Mowiol 4-88 mounting media (Sigma Aldrich). Microphotographs were obtained utilizing either Zeiss LSM 700 or 780 confocal microscopes.

**NMJ immunohistochemistry**. For muscle analysis and NMJ quantification, mice were anesthetized by Pentobarbital (250 mg/kg) and sacrificed by decapitation at three different timepoints: postnatal day 45, 63, and 84. *Gastrocnemius, tibialis anterior,* and *soleus* muscles were dissected and fixed in 4% PFA (Histolab) for 20 min, washed twice in PBS for 10 min, and cryoprotected in 30% sucrose. The muscles were then sectioned in a cryostat to obtain 100 μm thick slices which were collected in a 24-well plate (Falcon) for immunohistochemistry. Free-floating sections were blocked for 1 h in PBS-T and incubated for 48 h at 4 °C in primary antibodies diluted in blocking solution (Supplementary Table 1). Sections were then washed 1 h in PBS-T and incubated 3 h in secondary antibody (1:500, Alexa Fluor 555, A31570, Invitrogen) at room temperature. After three washes in PBS for 10 min at room temperature, sections were incubated with an α-Bungarotoxin Alexa Fluor 488 conjugated antibody for 10 min at room temperature. After two 5 min washes in PBS, sections were mounted on SuperFrost Plus slides, dried overnight, and cover slipped using Mowiol 4-88 mounting media. Microphotographs were obtained utilizing Zeiss LSM 700 confocal microscope.

**RNAscope® in situ hybridization**. Mice were anesthetized with an overdose of Pentobarbital (250 mg/kg) and sacrificed by decapitation. Timepoints were: postnatal day 45, 63, and 84 for analysis of En1 expression. For GIRK1/2 expression analysis, spinal cords were dissected one week after CNO administration. Spinal cords were dissected and snap frozen in isopentane kept in dry ice for cryoprotection. Coronal sections of the lumbar spinal cord were cut at 12 μm on a cryostat and collected on Superfrost Plus slides (Thermo Fisher Scientific). In situ hybridization was performed using the RNAscope® Multiplex Fluorescent Reagent Kit v2 Assay (Advanced Cell Diagnostics, Bio-techne). Pre-treatment and assay were performed as per the user manual, using the suggested HybEZ™ Oven for incubations. Samples were always stored overnight at room temperature in 5X Saline Sodium Citrate after probe hybridization. A ChAT probe was included as counterstaining in the En1 expression experiments. Catalogue numbers of all used probes are given in Supplementary Table 2. Opal™ 520 and 570 dyes (1:1500, FP1487001KT, FP1488001KT, Akoya Biosciences) were used as fluorophores. Counterstaining with NeuroTrace 640 (1:200 in PBS, N21483, Invitrogen) was performed prior to DAPI (provided in the V2 RNAscope kit) staining by 2 h incubation at room temperature followed by two washes in PBS. Microphotographs were obtained utilizing a Zeiss LSM 900 confocal microscope.

**Image analysis**. All immunohistochemistry image analysis and quantifications were conducted utilizing Fiji software. For synaptic density quantifications, microphotographs, obtained with a 20× objective – zoom = 1, were transformed to a grayscale 8-bit images and quantified after applying a threshold for signal

intensity/background correction. Images with a higher background/noise ratio were excluded. ROI were drawn around the motor neurons of interest that showed clear soma staining and visible nucleus and the pixel contained in the area of interest were quantified with the Fiji function Analyze particles. Masks of all quantified images were created to assure careful inclusion of pixels (Fig. 1). Between 8 and 10 images per mouse per condition were analyzed. For NMJ quantifications z-stack confocal images were acquired and the maximal projection was used for quantification. Microphotographs were acquired with 20× objective – zoom = 1 and 16 pictures per mouse per condition were quantified. Depending on the level of innervation, neuromuscular junctions (NMJs) were categorized as full, partial or empty (Fig. 2).

For in situ hybridization analysis, microphotographs were obtained using a 20× objective – zoom = 1. For each sample, $3 \times 3$ tiled images of each hemisection were acquired for a total of 8–10 tiled images per mouse per condition. Quantification of En1 positive cells was performed manually by two independent experimenters using ZEN Blue software (Zeiss). Neurons positive for En1 and located in the ventro-medial area of the spinal cord were included in the analysis. All the quantified En1 positive neurons presented clear NeuroTrace and DAPI counter staining. For all En1 neurons, area, perimeter, and signal intensity were analyzed and intensity averaged by the area was used for quantification of transcript expression. Percentages of cell quantifications and intensities were normalized utilizing values obtained from age-matching wild-type littermates.

**DigiGait treadmill test**. Locomotor performance was assessed using the DigiGait motorized transparent treadmill (Mouse Specifics, Inc.), which allows recording of animals from both a ventral and lateral view. Mice were pre-trained on the treadmill at speeds of 15 and 20 cm/s at postnatal day 42 and then tested weekly from postnatal day 49–112 for analysis of disease progression. For inhibitory chemogenetic experiments (activation of inhibitory DREADD receptors), age-matching mice were tested before and after administration of clozapine-N-oxide (CN0). After a 2 min acclimatization to the treadmill, mice were recorded at a speed of 20 cm/s for 10 s in three consecutive trials with 2 min rest periods in between recordings. Belt speed was adjusted to 15, 10, or 5 cm/s as needed, depending on the locomotor capability.

**Digital tracking analysis**. The videos captured during the treadmill experiments were analyzed in DeepLabCut (DLC)[48] to extract tracks based on digital markers placed on the mice (Figs. 5a and 8a). For the ventral view eleven digital markers were placed on the mice in a total of 50 frames. These labeled frames extracted from a subset of the videos created the training labels used for automatic tracking in DLC. A pre-trained ResNet-50[69] model provided in DLC was further fine-tuned with these labels and trained for a total of 4500 iterations. This fine-tuned model was then used to predict tracks from all the videos.

The tracks output by DLC were further analyzed to extract the locomotor measures reported in the manuscript. Details of estimating the reported measures are described below.

We extracted the instantaneous horizontal displacement of the mice $x_t$ on the treadmill, collected them into a vector denoted

$$\mathbf{X}^i = [x_1, ..., x_K] \qquad (1)$$

where $t = [1, .., K]$ and $K$ is the total number of frames in the video. Individual mice are indexed with superscript $i$, which is dropped in the remaining description for ease of notation.

*Speed*. Position estimates from all the markers, except the four used to track the paws, were utilised to estimate instantaneous speed of the mice. Difference in position, $\Delta x_t$, between the tracked markers in successive frames and the inter-frame duration, $\Delta t$, were used to estimate the instantaneous speed per tracked marker:

$$v_m = \frac{\Delta x_t}{\Delta t}(\text{cms}^{-1}) \text{ where } m = [1, ..., 7] \qquad (2)$$

The estimates from the seven markers were used to obtain the average instantaneous speed at time $t$ in the video, $v_t$:

$$v_t = \frac{1}{7}\sum_{m=1}^{7} v_m (\text{cms}^{-1}) \qquad (3)$$

A moving average filter of window size 10 was used to reduce the high-frequency noise in the speed estimates. The average speed per video was estimated by computing the mean over the instantaneous speed throughout the video:

$$v = \frac{1}{N}\sum_{t=1}^{K} v_t (\text{cms}^{-1}) \qquad (4)$$

*Acceleration*. Sudden increase or decrease in the speed of the mice were captured by estimating the instantaneous acceleration derived from the instantaneous speed estimates:

$$a_t = \frac{\Delta v_t}{\Delta t}(\text{cms}^{-2}) \qquad (5)$$

We set a threshold of 0.25 s of continuous deceleration to detect the event as a drag and conversely a duration of 0.25 s of continuous acceleration as a recovery

event. The maximum acceleration attained by the mice in each video is reported as peak acceleration:

$$a_P = \max(a_t)(\text{cms}^{-2}) \qquad (6)$$

These acceleration measures are reported in Fig. 4b, e, f.

Coordination: The tracks for the four paws were used to estimate the stride length and step frequency of the mice. We first estimated the total number of steps, $S$, in the video by analysing the relative position between the left and the right hindlimbs (Fig. 3-i). This provided an estimate of the step frequency:

$$F = \frac{S}{T}(\text{Hz}) \qquad (7)$$

where $T$ is the video duration.

Stride length, $L$ was estimated from the average speed and step frequency:

$$L = vF(\text{cm}) \qquad (8)$$

For each step, the phase difference between the left and right limbs, $\Phi_s$ where $s = [1, ..., S]$, were computed from the cadence plot in (Fig. 3I). The individual phase values were then plotted on a circle representing the interval of possible phases from 0 to 360 degrees. The average phase difference over all the steps per mouse, $\Phi$, were computed using circular statistics[70]. The average phase is indicated by the direction of the vector originating from the center of the circle. Phase values around 180 degrees indicate alternation while phase values around 0 (or 360) degrees indicate synchrony. The individual mouse level coordination profiles were further aggregated over all mice within a study group to report the group level coordination as seen in all the circular plots (Figs. 4g, 5h, 6i).

**Lateral kinematics**. The lateral view videos were first analyzed with DeepLabCut to obtain tracks based on six digital markers (corresponding to iliac crest, hip, knee, ankle, foot, and toe) on the visible left hindlimb (Fig. 8a). These tracked points were used to compute the instantaneous joint angles (hip-, knee-, ankle-, foot-angles) as the angle between the vectors formed by any three consecutive markers under consideration. For instance, the hip angle was computed as the angle between the vector connecting the hip and crest with the vector connecting the hip and knee. The difference between the maximum and minimum angles within each step cycle was estimated to report the average joint angles per animal in Fig. 8. A full step cycle was assumed when the paw of the mouse was in contact with the belt (stance), raised above (swing), and lowered back on to the belt. We used 15 complete cycles per animal to compute the average difference in the maximum and minimum joint angles. The stick diagram visualization of the lateral kinematics was derived from the tracks obtained from DeepLabCut. Stance and swing phases are indicated based on the contact of the foot with the belt of the treadmill.

**Grip strength test**. Low force motor performance was assessed by measuring hindlimb grip strength. Measurements of the hindpaw strength were performed by holding the mouse gently allowing the animal to grab on the metal bar by the hindpaws only[71]. A computerized grip strength meter (Cat. No. 47200, Ugo Basile) was used. SOD1[G93A] mice were tested at postnatal days 45, 60, 90, 120, and 140 for analysis of disease progression, as well as before and 10–15 min after administration of CNO for analysis of chemogenetic inhibition of En1 spinal interneurons. The peak force was recorded in gram-force (gf) and normalized to body mass. All measurements were performed in triplicates.

**Chemogenetic experiments**. For experiments utilizing inhibitory DREADDs, both males and females En1[cre];HoxB8[FlipO];RC::Di mice between postnatal day 63 and 90 were used. Mice were previously trained in the treadmill at 15 and 20 cm/s speeds and then tested as described above before and after injection. Also for the grip strength test, animals were previously trained and then tested before and after CNO delivery. Clozapine-N-oxide (CNO, Tocris, #4936) was diluted in saline at a concentration of 1 mg/kg and delivered intraperitoneally. Mice were then tested every 5 min after injections up to 20–30 min.

**Statistics**. All statistical analysis was performed with GraphPad software unless otherwise specified. No statistical methods were used to pre-determine sample sizes; our sample sizes are similar to those reported in the previous publications[64,54,72]. Mice were randomly allocated to different groups for the in vivo experiments using a block design. Data collection was performed blind to the conditions of the experiments although, in behavioral assessment, ALS mice at later stages could easily be recognized. Normal distribution of the populations was assumed but not formally tested. All tested mice were included in the study, apart from one SOD1[G93A] excluded from the survival assessment due to the appearance of wounds that required sacrifice before humane end-stage was reached. Unpaired Student's $t$ test was used to compare two groups and one-way ANOVA was used for multiple comparisons. When taking into account multiple variables, two-way ANOVA was used. ANOVA tests were followed by appropriate post hoc analysis as indicated in the figure legends. All results are expressed as mean ± SEM, reported n values represent distinct biological replicates. $P < 0.05$ was considered statistically significant, asterisks in figures and figure legends are *$P < 0.05$, **$P < 0.01$ or ***$P < 0.001$. To compare the left-right coordination during disease progression and

before and after treatment with CNO we used Watson–Williams test using 15 steps per mouse.

**Reporting summary**. Further information on research design is available in the Nature Research Reporting Summary linked to this article.

## Data availability

The data that support the findings of this study are available from the corresponding authors upon reasonable request. Source data are provided with this paper.

## Code availability

The code used to analyze data and produce figure content is available at https://github.com/kiehnlab/Locomotor-Allodi2021[73]

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

## Acknowledgements

We thank Dr. Susan Dymecki for providing the conditional RC::FPDi mouse. We acknowledge the Core Facility for Integrated Microscopy, Faculty of Health and Medical Sciences, University of Copenhagen and the Department of Experimental Medicine, University of Copenhagen, especially Carina Nyberg and Pernille Callesen. We thank Iryna Vesth-Hansen for technical assistance, and members of the Kiehn lab for discussion and comments on previous versions of this manuscript. The authors also thank Peter Simonsson for the drawing used in Figs. 3, 7, and 9. This work was supported by the Lundbeck Foundation (I.A.), the Björklund foundation (I.A.), the A.P. Møller foundation (I.A.), the Novo Nordisk Laureate Program (O.K., NNF15OC0014186), The Lundbeck Foundation (O.K.), the Louis-Hansen foundation (R.M.R.) and The Faculty of Health and Medical Sciences (O.K.).

## Author contributions

Conceptualization I.A. and O.K.; Methodology I.A., R.M.R., R.S., P.L. and O.K.; Investigation: I.A. and R.M.R.; Writing – Original Draft, I.A. and O.K.; Supervision O.K.; Funding Acquisition, I.A. and O.K.

## Competing interests

The authors declare no competing interests.
