## [Peer Review File · Nature Communications]

REVIEWER COMMENTS

Reviewer #1 (Remarks to the Author):

In the present manuscript, the Authors aimed at defining the role of V1 glycinergic interneurons in shaping disease progression in ALS by using the SOD1G93A mouse model of the disease. They report that glycine-releasing interneurons differently impinge on fast-twitch fatigable fiber-innervating motor neurons and on motor neurons innervating slow-twitch resistant fibers. Moreover, V1 glycine synapses selectively retract from fast motor neurons during disease progression, paralleling the decrease of motor abilities. Finally, silencing of spinal V1 neurons in WT mice mimics SOD1G93A locomotor deficits and, most relevant, silencing of V1 neurons in SOD1G93A mice did not produced any further worsening of disease motor marks. The Authors conclude for a pivotal role of V1 inhibitory interneurons in determining fast motor neuron vulnerability in ALS. Based on the measure of MN death at an early symptomatic phase of the disease, they deduced that this occurrence represents a precocious event, preceding MN degeneration. The study is novel and generally well performed, methods are sound and clearly reported; however, I feel that a number of point should be met to improve the manuscript.

One major concern involves the statement that fast and slow MNs can be uniquely identified on the basis of MMP9 or ErrBeta labelling, respectively. There is no literature support for this assumption throughout the manuscript. The Authors should report the appropriate evidences in the literature

The same consideration may apply to V1 interneuron. Does EN1 selectively label all V1 interneurons and at every specific maturation stage? The appropriate literature should be quoted.

Papers using the same animal model show that MN death occurs well in advance of the results in the manuscript (no significant changes at 84 days of life). This is relevant, being at the basis of the assumption that the loss of glycine innervation precedes MN degeneration. Oddly, these data are reported in the supplementary material, in non-ad hoc experiments. This issue should be addressed more in deep, to confirm this difference and determine the onset of motor neuron loss.

Points

1. Page 3 bottom and page 4 top: please quote the relevant references (see above)
2. Page 5, lines 8-9: mouse survival average is not commonly used to define survival probability; Kaplan Meyer representation is the gold standard. Additionally, the number of animals, especially of controls, is definitively too low for this estimation.
3. Page 5, lines 14-15: text refer to S Fig. 1C-E, but only WT and double mutant mouse values are reported therein. Data of SODG93A mice should be introduced for the correct comparison. Instead, comparison between WT and double mutants is correct for the assumptions at lines 4-5 from bottom.
4. Page 6, lines 6-9 from bottom: text is confusing. SFig. 2CD refer to copy number and weight not to survival. Moreover, weight difference between WT and transgenic mice is significant at 120 days only and there is not a decline of values (several authors consider the time at which weight decreases as an indication of the disease onset); the same applies to SFig. 1. Survival average values are reported in the text only for SOD1G93A-En1-Cre mice (as an alternative, the two genotypes share the same mean and standard error).
5. Page 8, lines 15-21: data reported in S Fig. 3 are relevant for the main story and should

be displayed as a regular figure.

Minor point

- a. page 4, line 6: found for find
- b. page 4, line 4 from bottom: the synaptophysin abbreviation should be indicated (SYN)
- c. page 5 line 6: use SYN here (the same at page 5, line 13 and line 3 from bottom)
- d. Page 7, line 2: please indicate that the data reported in the previous text are shown in Fig 2E
- e. Page 9, lines 19-20: Authors claim that all animals showed reduced speed but, from Fig. 5B, four of them acted at almost the same speed
- f. Fig. 1, panels K and L: the use of different colors should help the reader to distinguish the different time points
- g. SFig. 1: lower panels should be classified as C, D (not B) and, the following panels as E and F. Moreover, the absolute number of MNs should be indicated, at least in the legend to the figure

Reviewer #2 (Remarks to the Author):

The MS by Allodi and co-authors reports the deficit in V1-motoneuron connectivity in SOD^{G93A}-ALS mice models prior to motoneuron degeneration and prior to significant muscle denervation. Such changes in spinal inhibitory circuits, which might contribute to spinal-microcircuit mediated motoneuron vulnerability in ALS, suggest the presence of an interneuronal affection that may initiate ALS and lead to functional, i.e., locomotor, symptoms at relatively early stages of ALS disease.

The work presents a large amount of elegant experiments, is well illustrated and nicely written. The results are novel and highly relevant, the authors' claims are well sustained by the evidences shown which combine histological, functional and genetic manipulation tools. I have listed some minor comments below.

Introduction page 3: in the second paragraph, the sentence "In Sod1 mice the SOD1-induced motor neuron degeneration has been linked to motor neuron specific toxicity as well as non-motor...." is very long and slightly verbose, I suggest to shorten it or split in two sentences, as it is now the reading is tiresome and confusing.

Results: the authors often refer to "retraction" of gly V1 synapses from MN (FF) soma, I am intrigued by the suggested mechanism(s) of such retraction, since this could be a therapeutic target, please try to discuss this issue and the potential use of markers of retraction of synapses. Do the authors imply that synapses are eliminated or translocated?

Results: at page 7 in the second paragraph, the first sentence is not clear to me (starting with "Since fast and slow motor neurons are recruited in different motor tasks -...."). Why testing SOD1 mice at different postnatal times correlate to recruitment of fast and slow MN in different tasks?

The loss of En1+ terminals on fast motoneuron is well documented, is there also a loss of V1 neurons *per se* in SOD1^{G93A}? It is really a sort of *synaptopathy*, with loss/retraction of synapses? The occlusion experiments would have similar results in case of interneuron loss, correct?

Discussion: at page 12 bottom, the discussion on the dragging and denervation is slightly confusing: the onset of locomotor phenotype is between p49-p63, while clear denervation appear afterwards, as also stated by the authors (page 8 of results for ex) and supported by the strength analysis, thus given, how comes that the dragging is due to muscle denervation?

I am aware of page and words limits, but I would find beneficial to discuss different mice models of ALS, in particular in view of early locomotor symptoms. SOD1 are among the most widely used ALS models, but are not the only one and are thought to mostly target the familiar form of the disease. It would be interesting to assess whether in diverse models a similar early interneuronal deficit is observed.

Reviewer #3 (Remarks to the Author):

In the current manuscript, Allodi et al reported that spinal V1 interneurons dis-proportionally innervate fast and slow motor neurons (MNs). The loss of V1 synaptic terminals on fast MNs in SOD1G93A mice, an animal model for ALS disease, paralleled their development of 'locomotor phenotypes', both of which occurred earlier than the degeneration of MNs and the traditional onset point of ALS-like symptoms. In addition, the reversible suppression of V1 neuronal activity resembled the early locomotor phenotypes in SOD1G93A mice. Therefore, the authors claimed that the spinal V1 INs could be a 'source of motor neuronal vulnerability in ALS'. Previous studies in human and different animal models have noted the decrease in the ratio of inhibitory/excitatory synapses, specifically in MNs of ALS patients or animals. V1 INs are one of the major inhibitory IN populations that can directly innervate MNs in the spinal cord, and so it may not be surprising that V1 INs would contribute to an early imbalance of inhibitory and excitatory inputs to MNs. More interestingly, V1 mutants did show slower locomotor movement. Therefore, it would be very important and exciting to truly confirm experimentally that V1 INs are the major player in the early development of ALS. Disease. Unfortunately, however, some experiments in the current manuscript were not very carefully executed, and some claims here were not very convincing. My main concerns are the following.

1. Quantification of synaptic density. First, the quality of the immuno-staining in Fig1, 2 was quite poor. I am not sure if any of the counts are trustworthy, judging by the representative images showed in the figures. As a matter of fact, both MMP9 and ErrBeta antibodies can be quite tricky. It might be helpful to first show a group of neurons with separate channels to show different antibody staining, and then focus on individual neurons. It is even hard to judge whether the fluorescent signals of the antibody staining were specific in their experiments. Second, it is not clear how they determined the surface of the MNs. The size and surface area of MNs can vary a lot. Since fast MNs are generally bigger than slow MNs, shouldn't the total number of synapses on fast MNs be larger as well? This wouldn't necessarily mean the density of the synapses is larger. Third, it would be helpful and useful to check the changes of excitatory synapses (VGLUT2, VChATs...) in parallel with SOD1 mutants as a reference.

2. The animal and viral models. It is surprising to me that they only tried the RosaYFP reporter line, and then turned to AAVs. There are many other reporter lines, such as ROSASyp/tdTom in JAX. Using AAVs in this case may not be the best choice. It has always

been a concern that postnatal expression of En1 in spinal neurons, at least some of them, may decrease. More importantly, it is not clear how En1 expression in different V1 INs changes in SOD1 mutant animals, which would greatly affect the expression of cre and SynEGFP. The low counts of GFP+ synapses might only reflect the decreased expression of En1 in SOD1G93A animals. In addition, we all know that viral injection and expression efficiency can vary dramatically from animal to animal, which adds another uncertainty to any quantificational analysis. Therefore, without further proper characterization and quality control of their AAV tools, I wouldn't consider the results in Fig 2 to be very useful.

3. The locomotor phenotypes. It is interesting that the authors discovered the 'locomotor phenotypes in the young SOD1G93A mice much earlier than the time when MNs start degenerating. More interestingly, they claimed slow movement in SOD1G93A mice at a young age similar to what was shown in V1 mutant animals. To me, however, these two slow movements could just be totally different phenotypes. 1) Various alternations of activities of different spinal interneurons can lead to low-speed movement. The gait analysis in the manuscript was quite general and vague. It is hard to judge whether these two are the same and pin down whether V1 INs are the main source of the SOD1 phenotype. Particularly, there were differences between SOD and V1 deficient mice even in their analysis. 2) To me, Fig S3 showed that NMJs in TA and GN have already started deteriorating at P45. The grabbing test could use different MN pools from the movement. 3) Even if we could agree that the slow movement of SOD1 mutants was caused by V1 defects, it would be a phenotype of V1 IN subpopulations in the core of central pattern generator (CPG) circuits, which might or might not have anything to do with pre-MN V1 interneurons. Therefore, the authors might want to first investigate whether the number or function of the general or subgroups of spinal V1 INs changed in SOD1 mutants. It seems that the authors are trying to mix issues from different layers of circuits together. Overall, I would argue that the locomotor phenotypes in young SOD1G93A mutants are the collective effects of many spinal neurons.

We would like to thank the reviewers for insightful comments and suggestions for further experiments to improve the study.

Reviewer #1 (Remarks to the Author):

In the present manuscript, the Authors aimed at defining the role of V1 glycinergic interneurons in shaping disease progression in ALS by using the SOD1G93A mouse model of the disease. They report that glycine-releasing interneurons differently impinge on fast-twitch fatigable fiber-innervating motor neurons and on motor neurons innervating slow-twitch resistant fibers. Moreover, V1 glycine synapses selectively retract from fast motor neurons during disease progression, paralleling the decrease of motor abilities. Finally, silencing of spinal V1 neurons in WT mice mimics SOD1G93A locomotor deficits and, most relevant, silencing of V1 neurons in SOD1G93A mice did not produced any further worsening of disease motor marks. The Authors conclude for a pivotal role of V1 inhibitory interneurons in determining fast motor neuron vulnerability in ALS. Based on the measure of MN death at an early symptomatic phase of the disease, they deduced that this occurrence represents a precocious event, preceding MN degeneration. The study is novel and generally well performed, methods are sound and clearly reported; however, I feel that a number of point should be met to improve the manuscript.

We thank reviewer #1 for these positive comments and support of the study.

One major concern involves the statement that fast and slow MNs can be uniquely identified on the basis of MMP9 or ErrBeta labelling, respectively. There is no literature support for this assumption throughout the manuscript. The Authors should report the appropriate evidences in the literature

We thank the reviewer for pointing out that there was no literature references to support this assumption. There is a number of studies where the evidences have been reported. We are now referring to these studies in line 88-89, page 4 of the manuscript and we report them here as well:

matrix metalloproteinase-9 (MMP-9): Kaplan et al. (2014). Neuronal matrix metalloproteinase-9 is a determinant of selective neurodegeneration. *Neuron* 81(2):333-48.

Additional references utilizing MMP-9 as a marker for FF MNs:

-*Bernard-Marissal N et al (2015) Dysfunction in endoplasmic reticulum-mitochondria crosstalk underlies SIGMAR1 loss of function mediated motor neuron degeneration. Brain* 138(4): 875-890

-*Spiller KJ et al (2016) Selective Motor Neuron Resistance and Recovery in a New Inducible Mouse Model of TDP-43 Proteinopathy. J Neurosci* 36(29):7707–7717

-Morisaki Y et al (2016) Selective Expression of Osteopontin in ALS-resistant Motor Neurons is a Critical Determinant of Late Phase Neurodegeneration Mediated by Matrix Metalloproteinase-9. *Scientific Reports* 6, 27354

-Kelley KW et al (2018) Kir4.1-Dependent Astrocyte-Fast Motor Neuron Interactions Are Required for Peak Strength. *Neuron* 98, 1-14

-Rozas R et al (2021) Protein disulfide isomerase ERp57 protects early muscle denervation in experimental ALS. *Acta Neuropathol Commun* 9(21)

ErrBeta: Enjin et al. (2010). Identification of novel spinal cholinergic genetic subtypes disclose *Chodl* and *Pitx2* as markers for fast motor neurons and partition cells. *Journal of Comparative Neurology* 518(12):2284-304.

Additional references utilizing ErrBeta as a marker for S MNs:

-Leroy F et al (2014) Early intrinsic hyperexcitability does not contribute to motoneuron degeneration in amyotrophic lateral sclerosis. *eLife* 3:e04046

The same consideration may apply to V1 interneuron. Does EN1 selectively label all V1 interneurons and at every specific maturation stage? The appropriate literature should be quoted.

The presence of the transcription factor En1 is what defines interneurons as V1 interneurons as first defined in embryonic spinal cord by

-Ericson et al (1997) Pax6 controls progenitor cell identity and neuronal fate in response to graded Shh signaling. *Cell* 90(1):169-80

And

-Matisse MP, Joyner AL (1997) Expression patterns of developmental control genes in normal and Engrailed-1 mutant mouse spinal cord reveal early diversity in developing interneurons. *J Neurosci* 17(20):7805-16

The V1 interneurons have been characterized extensively both pre and early postnatally (*Sapir T et al (2004) J Neurosci* 24:1255–1264; *Saueressig H et al (1999) Development* 126:4201–4212; *Bikoff JB et al (2016) Cell* 165, 207–219) and physiologically (*Wang Z et al (2008) J Neurophysiol* 100:185–196; *Gosgnach S et al (2006) Nature* 440, 215–219) in all cases defining V1 as expressing En1.

Studies by Goulding's group have also studied V1 interneurons in adult mice again defining them by expression of En1 (*Zhang J et al (2014) Neuron* 82(1):138-50; *Britz O et al (2015) eLife* 4:e04718). There is no other way to define the entire V1 population than by expression of En1. It is possible that En1 will be down-regulated in some neurons postnatally so that some En1-negative neurons now do not look like V1 interneurons anymore. No one including us have addressed this issue in detail, but we now show that En1 is indeed present late postnatally in many interneurons which we by definition call V1 interneurons. These data are

presented in new Figure 4 and paragraph “*The number of Engrailed 1 positive neurons is reduced in SOD1^{G93A} mouse compared to wild type littermates before motor neuron degeneration*” line 178.

Moreover, we use the original En1^{Cre} mouse from Joyner’s lab that was generated to characterize the V1 population. We are now referring to the appropriate literature in the paper related to characterization of En1 expression at line 63-65 of the Introduction paragraph, page 4.

Papers using the same animal model show that MN death occurs well in advance of the results in the manuscript (no significant changes at 84 days of life). This is relevant, being at the basis of the assumption that the loss of glycine innervation precedes MN degeneration. Oddly, these data are reported in the supplementary material, in non-ad hoc experiments. This issue should be addressed more in deep, to confirm this difference and determine the onset of motor neuron loss.

This point is well taken. We have now included one more data point (day 112) in the analysis in which we show significant motor neuron death compared to day 84 in the same mouse strain. As previously reported by studies performing Nissl staining in the same SOD1^{G93A} mouse model at P110-112 (*Fischer LR et al (2004) Experimental Neurology 185(2):232-240; Zhao W et al (2011) Molecular Neurodegeneration 6,51; Tan H et al (2020) Cell Death & Differentiation 27, 1369–1382*), we observe a clear motor neuron death at this time point (wt = 15.4±3.4, P84 = 12.1±0.7, P112 = 5.4±0.8). These data are now included in Supplementary Fig.2 and quantifications are included in the Figure legend as suggested below. We also show that there is no motor neuron shrinkage at day 84 compared to control mice. We have added a paragraph detailing these findings (page 5 lines 110-112).

Points

1. Page 3 bottom and page 4 top: please quote the relevant references (see above)

This is now done: line 88-89, page 4.

2. Page 5, lines 8-9: mouse survival average is not commonly used to define survival probability; Kaplan Meyer representation is the gold standard. Additionally, the number of animals, especially of controls, is definitively too low for this estimation.

We have followed this advice and added survival curve (Kaplan Meyer curve) for all conditions in manuscript. We have also increased the number of mice included in the survival curve (SOD1^{G93A} = 11; SOD1^{G93A}; GlyT2GFP = 8; SOD1^{G93A}; En1cre = 6; SOD1^{G93A};En1cre; HoxB8FlipO; RC::FPDi = 10). These data are reported in a new Supplementary Figure 1C.

3. Page 5, lines 14-15: text refer to S Fig. 1C-E, but only WT and double mutant mouse values are reported therein. Data of SODG93A mice should be introduced for the correct

comparison. Instead, comparison between WT and double mutants is correct for the assumptions at lines 4-5 from bottom.

The text has now been changed to compare SOD1^{G93A} vs SOD1^{G93A}; GlyT2 (line 108-109 page 5).

4. Page 6, lines 6-9 from bottom: text is confusing. SFig. 2CD refer to copy number and weight not to survival. Moreover, weight difference between WT and transgenic mice is significant at 120 days only and there is not a decline of values (several authors consider the time at which weight decreases as an indication of the disease onset); the same applies to SFig. 1. Survival average values are reported in the text only for SOD1G93A-En1-Cre mice (as an alternative, the two genotypes share the same mean and standard error).

The number of mice included in the graph has been increased and now reporting a significant difference in weight starting from day 90 and onwards (Fig. 2A). Moreover, comparisons of weights for each condition is now reported in the new Supplementary Figure 1A-B which includes all the numbers for male mice up to P112.

5. Page 8, lines 15-21: data reported in S Fig. 3 are relevant for the main story and should be displayed as a regular figure.

Thank you for this suggestion. The data in previous Fig S3 is moved to Fig. 2A-E.

Minor point

a. page 4, line 6: found for find:

Changed

b. page 4, line 4 from bottom: the synaptophysin abbreviation should be indicated (SYN)

Abbreviation is added.

c. page 5 line 6: use SYN here (the same at page 5, line 13 and line 3 from bottom)

Changed from SYP to SYN in the whole manuscript.

d. Page 7, line 2: please indicate that the data reported in the previous text are shown in Fig 2E

Done, but now the same data is shown in S Fig. 3E as indicated at line 170, page 7.

e. Page 9, lines 19-20: Authors claim that all animals showed reduced speed but, from Fig. 5B, four of them acted at almost the same speed

This is correct and we point out this more clearly in the text. Phrasing is changed to “the majority”.

f. Fig. 1, panels K and L: the use of different colors should help the reader to distinguish the different time points

Thanks for this suggestion. Different colors are chosen.

g. SFig. 1: lower panels should be classified as C, D (not B) and, the following panels as E and F. Moreover, the absolute number of MNs should be indicated, at least in the legend to the figure

This figure has been changed and is now Supplementary Fig. 2. The absolute number of motor neurons is now indicated in the legend as suggested.

Reviewer #2 (Remarks to the Author):

The MS by Allodi and co-authors reports the deficit in V1-motoneuron connectivity in SOD^{G93A}-ALS mice models prior to motoneuron degeneration and prior to significant muscle denervation. Such changes in spinal inhibitory circuits, which might contribute to spinal-microcircuit mediated motoneuron vulnerability in ALS, suggest the presence of an interneuronal affection that may initiate ALS and lead to functional, i.e., locomotor, symptoms at relatively early stages of ALS disease. The work presents a large amount of elegant experiments, is well illustrated and nicely written. The results are novel and highly relevant, the authors' claims are well sustained by the evidences shown which combine histological, functional and genetic manipulation tools.

I have listed some minor comments below.

We thank the reviewer for these positive comments.

Introduction page 3: in the second paragraph, the sentence "In Sod1 mice the SOD1-induced motor neuron degeneration has been linked to motor neuron specific toxicity as well as non-motor...." is very long and slightly verbose, I suggest to shorten it or split in two sentences, as it is now the reading is tiresome and confusing.

The sentence has been shortened as suggested.

Results: the authors often refer to "retraction" of gly V1 synapses from MN (FF) soma, I am intrigued by the suggested mechanism(s) of such retraction, since this could be a therapeutic target, please try to discuss this issue and the potential use of markers of retraction of synapses. Do the authors imply that synapses are eliminated or translocated?

Thanks for pointing out this. We have chosen the word retraction because we think that the synapses are indeed eliminated from motor neurons. The fact that there are overall fewer En1 synapses in the spinal cord of the SOD1 mice suggests that the synapses are not translocated although we do not have any means to show that directly. We agree that the retraction offers a therapeutic target that we are going to address in future studies. We now discuss these issues more deeply in the Discussion, page 14 line 383-391.

Results: at page 7 in the second paragraph, the first sentence is not clear to me (starting with "Since fast and slow motor neurons are recruited in different motor tasks -..."). Why testing SOD1 mice at different postnatal times correlate to recruitment of fast and slow MN in different tasks?

We apologize for not being clear – the point is that fast and slow motor neurons may be recruited in different motor tasks (tonic and dynamic) not at different postnatal times. We have changed the sentence in line 199-203 page 8.

The loss of En1+ terminals on fast motoneuron is well documented, is there also a loss of V1 neurons *per se* in SOD1^{G93A}? It is really a sort of *synaptopathy*, with loss/retraction of synapses? The occlusion experiments would have similar results increase of interneuron loss, correct?

We have addressed this point separately now in new experiments evaluating the number of En1 positive cells in the spinal cord using RNAscope *in situ* hybridization and En1 transcripts at time points P45-P63-P84. This new data set is now included in a new Figure 2A-E. There is a significant drop in the number of En1 positive cells in the ventral horn to about 75% of control values starting at P63. This drop is comparably smaller than the synaptic retraction of En1 synapses from motor neurons which would suggest a synaptopathy. An alternative explanation is that the lost En1 neurons could be the ones that specifically innervate fast motor neurons while the ones innervating slow motor neurons remain healthy. We are unable to define an experiment that will address these possibilities directly. Transsynaptic labeling e.g, while able to label specific pre-motor circuits synaptic neurons would not be able to differentiate loss of synapse or dying neurons: in both cases the number of transsynaptic labelled cells would decrease.

Discussion: at page 12 bottom, the discussion on the dragging and denervation is slightly confusing: the onset of locomotor phenotype is between p49-p63, while clear denervation appear afterwards, as also stated by the authors (page 8 of results for ex) and supported by the strength analysis, thus given this, how comes that the dragging is due to muscle denervation?

This is a good point. We have changed the text and removed this confusion.

I am aware of page and words limits, but I would find beneficial to discuss different mice models of ALS, in particular in view of early locomotor symptoms. SOD1 are among the most widely used ALS models, but are not the only one and are thought to mostly target the familiar form of the disease. It would be interesting to assess whether in diverse models a similar early interneuronal deficit is observed.

We thank the reviewer for this suggestion. One of the future directions of our work is indeed to address experimentally changes in other ALS models as well, however, the SOD1^{G93A} strain is the only one available in our laboratory at the moment. We now discuss the possibilities and implications in the Discussion paragraph.

Reviewer #3 (Remarks to the Author):

In the current manuscript, Allodi et al reported that spinal V1 interneurons dis-proportionally innervate fast and slow motor neurons (MNs). The loss of V1 synaptic terminals on fast MNs in SOD1G93A mice, an animal model for ALS disease, paralleled their development of 'locomotor phenotypes', both of which occurred earlier than the degeneration of MNs and the traditional onset point of ALS-like symptoms. In addition, the reversible suppression of V1 neuronal activity resembled the early locomotor phenotypes in SOD1G93A mice. Therefore, the authors claimed that the spinal V1 INs could be a 'source of motor neuronal vulnerability in ALS'. Previous studies in human and different animal models have noted the decrease in the ratio of inhibitory/excitatory synapses, specifically in MNs of ALS patients or animals. V1 INs are one of the major inhibitory IN populations that can directly innervate MNs in the spinal cord, and so it may not be surprising that V1 INs would contribute to an early imbalance of inhibitory and excitatory inputs to MNs. More interestingly, V1 mutants did show slower locomotor movement. Therefore, it would be very important and exciting to truly confirm experimentally that V1 INs are the major player in the early development of ALS. Disease. Unfortunately, however, some experiments in the current manuscript were not very carefully executed, and some claims here were not very convincing. My main concerns are the following.

We are happy that the referee acknowledge that the findings are important and respectfully disagree that the experiments was not carefully executed. We have however performed more experiments to address several of the issues raised.

1. Quantification of synaptic density. First, the quality of the immuno-staining in Fig1, 2 was quite poor. I am not sure if any of the counts are trustworthy, judging by the representative images showed in the figures. As a matter of fact, both MMP9 and ErrBeta antibodies can be quite tricky. It might be helpful to first show a group of neurons with separate channels to show different antibody staining, and then focus on individual neurons. It is even hard to judge whether the fluorescent signals of the antibody staining were specific in their experiments.

We find the critique "I am not sure if any of the counts are trustworthy" unjustified and unfair. We have established protocols for MMP9 and ErrBeta antibody staining and they have been used by us and others, see also answer to reviewer #1 (ErrBeta: *Enjin et al (2010) J Comp Neurol* 518(12):2284-304; *Leroy F et al (2014) eLife* 3:e04046; MMP9: *Kaplan et al (2014) Neuron* 81(2):333-48; *Bernard-Marissal N et al (2015) Brain* 138(4): 875-890; *Spiller KJ et al (2016) J Neurosci* 36(29):7707-7717; *Morisaki Y et al (2016) Scientific Reports* 6, 27354; *Kelley KW et al (2018) Neuron* 98, 1-14). The GlyT2-GFP mouse has been thoroughly evaluated before and shown that GFP report glycinergic transmitter phenotype reliable in the spinal cord (*Restrepo E...Kiehn O (2009) J Comp Neurol* 517(2):177-92). The quantification of 'synapses' was

automatized and performed by two independent experimenters, moreover we only concentrated the quantification on soma-near 'synapses'. We are not aware that the quantification can be done differently. However, to improve the visual impression we now show the cells in separate channels for the different antibody staining as suggested in Figure 1.

Second, it is not clear how they determined the surface of the MNs. The size and surface area of MNs can vary a lot. Since fast MNs are generally bigger than slow MNs, shouldn't the total number of synapses on fast MNs be larger as well? This wouldn't necessarily mean the density of the synapses is larger.

The reviewer is absolutely correct, but we actually did that analysis in the first version of the manuscript and expressed synapses as density taking into account the size of motor neurons. We have made this clearer in the text to avoid confusion Page 5 line 95-96, Page 7 line 163-164, as well as in the Material and Methods *Image Analysis* section. Area and perimeter were analyzed for all motor neurons included in the analysis to assure that differences in synaptic densities were not due to changes in cell morphology, e.g. shrinkage (no shrinkage as shown in Supplementary Fig. 2).

Third, it would be helpful and useful to check the changes of excitatory synapses (VGLUT2, VChATs...) in parallel with SOD1 mutants as a reference.

Although this was not the scope of our study, we have now added new studies to investigate possible changes in excitatory glutamatergic interneuron synapses that are Vglut2 positive around motor neuron somata at P45 and P63 time points. This new data-set is now included in Supplementary Figure 2E-J. We cannot evaluate these synapses on dendrites because there is no way to assign dendrites to specific motor neuron pools. Notably we did not find significant changes in the number of Vglut2 positive synapses onto fast and slow motor neurons at these earlier timepoints.

We now also refer to these data in the discussion – page 15, line 394-395.

We did not evaluate the VaChT terminals. The only VaChT terminals from interneurons ending on motor neurons are C-Boutons originating in Pitx2 positive neurons close to the central canal. The changes in C-Boutons have been extensively studied in SOD1^{G93A} showing both increase and decrease in volume and number. A recent study evaluated all studies and provided new experiments and concluded that there are no differences in C-bouton size or numbers during disease progression (*Dukipati et al (2017) ENEURO.0281-16*). We therefore abstain from doing further studies of VaChT.

2. The animal and viral models. It is surprising to me that they only tried the RosaYFP reporter line, and then turned to AAVs. There are many other reporter lines, such a ROSASyp/tdTom in JAX. Using AAVs in this case may not be the best choice. It has always

been a concern that postnatal expression of En1 in spinal neurons, at least some of them, may decrease. More importantly, it is not clear how En1 expression in different V1 Ins changes in SOD1 mutant animals, which would greatly affect the expression of cre and SynEGFP. The low counts of GFP+ synapses might only reflect the decreased expression of En1 in SOD1^{G93A} animals. In addition, we all know that viral injection and expression efficiency can vary dramatically from animal to animal, which adds another uncertainty to any quantificational analysis. Therefore, without further proper characterization and quality control of their AAV tools, I wouldn't consider the results in Fig 2 to be very useful.

The reviewer states that using AAVs to visualize terminals may not be the best choice and that we should use another reporter line such as ROSASyp/tdTom to label terminals in the En-1 line. Unfortunately, since the ROSASyp/tdTom is not available in our laboratory, switching to ROSASyp/tdTom/En1/SOD1 crossing might seem trivial but it requires to generate new triple transgenic mice and to redo all these studies again from scratch with a timeline of more than a year. Moreover, using the transgenic approach does not address whether En1 expression in different V1 interneuron changes in SOD1 mutant mice as suggested by the reviewer.

We have therefore maintained the viral approach which we think is optimal to address the issue and provide very strong labelling of terminals. It is true that viral injections and expression efficiency can vary from animal to animal. However, we actually took this into account in our analysis already in the first version of our manuscript. We normalized the number of synaptic terminals to the number of labelled neurons (as described in page 7 line 160-162) which should alleviate confounding factors as variable injection sizes and variable expression efficiency. We have now made this point even clearer in the new version of the manuscript.

We now also include quantifications of En1 positive neurons as well as En1 transcript in control and SOD1 animals at three different time points, in order to include an investigation of the fate of the En1 population during disease progression. The first observation is that in wild-type mice there is a large cohort of En1 positive cells present in ventral spinal cord at the time points we investigate. This suggests that the down-regulation of En1 postnatally may be less than has been anticipated, and establishes that targets for the viral vector are present at this time point. Changes between controls and SOD1 mice is therefore likely to be due to the neurodegeneration progressing in the SOD1^{G93A} mice. Indeed, there is a significant drop in the number of En1 positive cells in the ventral horn to about 75% of control values starting at P63, which is further reduced to 68% at P84. This drop is comparably smaller than the synaptic retraction of En1 synapses from motor neurons which suggests that there is a specific retraction of synapses from fast motor neurons. We do acknowledge though that the lost En1 neurons could be the ones that specifically innervate fast motor neurons while the ones innervating slow motor neurons remain healthy. We are unable to define an experiment that

will address this directly. Transsynaptic labeling e.g. would not be able to differentiate loss of synapse or dying neurons: in both cases the number of transsynaptic labelled cells will decrease.

3. The locomotor phenotypes. It is interesting that the authors discovered the 'locomotor phenotypes in the young SOD1^{G93A} mice much earlier than the time when MNs start degenerating. More interestingly, they claimed slow movement in SOD1^{G93A} mice at a young age similar to what was shown in V1 mutant animals. To me, however, these two slow movements could just be totally different phenotypes. 1) Various alternations of activities of different spinal interneurons can lead to low-speed movement. The gait analysis in the manuscript was quite general and vague. It is hard to judge whether these two are the same and pin down whether V1 INs are the main source of the SOD1 phenotype. Particularly, there were differences between SOD and V1 deficient mice even in their analysis. 2) To me, Fig S3 showed that NMJs in TA and GN have already started deteriorating at P45. The grabbing test could use different MN pools from the movement. 3) Even if we could agree that the slow movement of SOD1 mutants was caused by V1 defects, it would be a phenotype of V1 IN subpopulations in the core of central pattern generator (CPG) circuits, which might or might not have anything to do with pre-MN V1 interneurons. Therefore, the authors might want to first investigate whether the number or function of the general or subgroups of spinal V1 INs changed in SOD1 mutants. It seems that the authors are trying to mix issues from different layers of circuits together. Overall, I would argue that the locomotor phenotypes in young SOD1^{G93A} mutants are the collective effects of many spinal neurons.

We appreciate the reviewer's arguments and respect his/her point of view. The reviewer state that gait analysis in the manuscript was quite general and vague but does not give any directions for what should be done to strengthen the analysis. However, using unbiased DeepLabCut analysis we have now added analysis of the four limbs when locomoting on the treadmill for all the different conditions which show consistent differences between SOD1^{G93A} and En1cre; HoxB8FlpO; RC::FPDi after CNO groups (now shown in the new Supplementary Figure 6). Here, homolateral coordination is shifted when compared to pre-symptomatic SOD1^{G93A} as well as in En1cre; HoxB8FlpO; RC::FPDi before CNO administration. Diagonal coordination of the limbs also remains unchanged. Concerning limb coordination at the segmental level, we show that there is no change in in left right coordination excluding affection of commissural neurons.

Moreover, a new cohort of animals (a total of 8 wt and 5 SOD1^{G93A} as well as 7 En1cre; HoxB8FlpO; RC::FPDi and 7 controls) was added to the study to perform a new analysis of intralimb kinematic from the lateral view. These data show a clear change in limb kinematic in SOD1^{G93A} and En1cre; HoxB8FlpO; RC::FPDi mice after CNO administration – with hyperflexion similar to what was found by Goulding's group after silencing of V1 neurons (Britz O et al 2015 eLife). These data are presented in new Figure 8.

The changes in speed of locomotion, step frequency and the stride length are similar to the phenotype that has been seen in both chronic and acute silencing of En1 neurons (*Gosgnash et al.* 2006 *Nature*; *Falgairolle M and O'Donovan MJ* 2019 *Plos Biology*). In particular, the hyperflexion seen in both SOD1^{G93A} and En1cre; HoxB8FlpO; RC::FPDi mice was also found by Goulding's group after silencing of V1 neurons (*Britz O et al* 2015 *eLife*). Moreover, we showed that En1 silencing has no effect in SOD1^{G93A} mice after "Onset of locomotor phenotype". Together, this new data and the one reported in the previous version of the manuscript support that the V1 neurons are contributing to the phenotype in SOD1^{G93A} mice. We find these data to be very strong.

In this point, the reviewer also suggests that we should investigate whether the number or function of the general or subgroups of spinal V1 interneurons changed in SOD1^{G93A} mutants. This is technically very difficult since there are more than 50 subtypes of V1 neuronal subpopulations (*Bikoff et al.* 2016 *Cell*) and few mice line, not generally available that allow us to target them. However, we are interested in addressing some of these issues in our future work.

REVIEWER COMMENTS

Reviewer #2 (Remarks to the Author):

The authors perform an accurate revision and thoroughly addressed all concerns, I have no further comments.

Reviewer #3 (Remarks to the Author):

The authors have well addressed my concerns. This is a much improved manuscript.